## Registered report

psychology/cognition

inequity aversion, social emotions, *Kinect*, fairness, body posture

**Author for correspondence:**
Stella C. Gerdemann
e-mail: stella.gerdemann@uni-leipzig.de

†Faculty of Education, Leipzig University, Jahnallee 59, 04109 Leipzig, Germany.

# The ontogeny of children's social emotions in response to (un)fairness

Stella C. Gerdemann[1,2,†], Katherine McAuliffe[3],
Peter R. Blake[4], Daniel B. M. Haun[5] and Robert Hepach[6]

[1]Department of Early Child Development, and [2]Leipzig Research Center for Early Child Development, Leipzig University, Leipzig, Germany
[3]Department of Psychology, Boston College, Boston, MA, USA
[4]Department of Psychological and Brain Sciences, Boston University, Boston, MA, USA
[5]Department of Comparative Cultural Psychology, Max Planck Institute for Evolutionary Anthropology, Leipzig, Germany
[6]Department of Experimental Psychology, University of Oxford, Oxford, UK

SCG, 0000-0002-7200-9499; PRB, 0000-0003-3968-9880

Humans have a deeply rooted sense of fairness, but its emotional foundation in early ontogeny remains poorly understood. Here, we asked if and when 4- to 10-year-old children show negative social emotions, such as shame or guilt, in response to advantageous unfairness expressed through a lowered body posture (measured using a *Kinect* depth sensor imaging camera). We found that older, but not younger children, showed more negative emotions, i.e. a reduced upper body posture, after unintentionally disadvantaging a peer on (4,1) trials than in response to fair (1,1) outcomes between themselves and others. Younger children, in contrast, expressed more negative emotions in response to the fair (1,1) split than in response to advantageous inequity. No systematic pattern of children's emotional responses was found in a non-social context, in which children divided resources between themselves and a non-social container. Supporting individual difference analyses showed that older children in the social context expressed negative emotions in response to advantageous inequity without directly acting on this negative emotional response by rejecting an advantageously unfair offer proposed by an experimenter at the end of the study. These findings shed new light on the emotional foundation of the human sense of fairness and suggest that children's negative emotional response to advantageous unfairness developmentally precedes their rejection of advantageously unfair resource distributions.

## 1. The ontogeny of children's social emotions in response to (un)fairness

A concern for fairness when distributing resources is considered a bedrock for human cooperation [1]. When need or effort between

individuals are similar, adults show *inequity aversion* [2], that is negative emotional and behavioural responses to unequal divisions of resources, both in the context of *disadvantageous inequity* when given comparably less than another, as well as in the case of *advantageous inequity* when given relatively more [3–7]. While disadvantageous resource distributions are rejected more often when a social partner receives more, disadvantageously unequal resource are also rejected in non-social contexts, when no recipient's stake is affected by the resource distribution [8,9]. This suggests that disadvantageous inequity aversion may be motivated by fairness concerns, as well as a non-social frustration over receiving less than one could. By contrast, advantageous inequity aversion requires resolving a conflict between non-social concerns (for gaining more than a fair share of a resource for oneself) and fairness norms (equitable distribution of resources), as these two motives are not aligned when receiving more than one deserves [10]. Social emotions, such as shame, guilt and pride, are theorized to have the evolved function of enabling humans to resolve such conflicts between self-interest and social norms, to the degree that they are internalized forms of an evaluation of oneself and one's behaviour as viewed from the perspective of one's social group (e.g. [11]). This raises the question if and when during ontogeny children show negative social emotions, such as shame or guilt, in response to advantageous inequity, and positive social emotions, such as pride, in response to fairness.

When self-interest is not at stake, in third-party contexts, 3-year-old children divide resources fairly between others, that is equally or according to merit [12,13]. Similarly, in first-party contexts, young children by age 4 show disadvantageous inequity aversion [8,14–16]. For instance, when resources are divided unequally between peers, such as through a (1,4) distribution, young children reject this distribution entirely and discard it into a trash bin [14]. Yet, advantageous inequity aversion, by contrast, does not appear to emerge until children are at least school-aged, and its development shows more variability across cultures than disadvantageous inequity aversion ([14,15,17–22] see [21] for a review). For instance, by age 7, children from US-American samples will rather throw an additional resource away, if receiving it would make an equal split unequal in favour of themselves [18,22]. In these studies children younger than age 7 appear either indifferent to the lack of fairness towards others or may prefer self-advantaging inequity. Thus, one study showed that 5- to 6-year-olds will take a cost to avoid fairness and disadvantage a peer, showing 'anti-equality' [16]. Given that children are aware of fairness norms by age 3 [23], it appears unlikely that young children's acceptance of advantageous inequity in many contexts is due to a lack of knowledge of fairness principles.

One issue with these studies is that there may be competing task demands that mask children's otherwise negative responses to advantageous inequity. Young children's acceptance of unfairness towards others may partially be the result of younger children's greater difficulty to inhibit a dominant response to accept large quantities of a resource, and thus may be the result of a lack of behavioural control ([24,25]; see also [23]). Moreover, in contexts in which resources are acquired collaboratively by peers without a proposed unequal resource distribution by an adult, young children divide resources fairly when doing so requires giving up an advantage ([26–28]; see also [29]). However, only older children reject advantageous resource distributions acquired through windfall [14,22]. Given the, in part, contradictory results across studies and contexts regarding children's behaviour in response to advantageous inequity, here, we systematically investigated the development of children's social emotions in response to (un)fairness.

Despite the relevance of social emotions, such as shame, guilt and pride, to following social norms and overriding one's self-interest [11], little empirical work has focused on the development of children's social emotions in response to (advantageous) (un)fairness. In one study, Kogut [30] found that children by fourth grade, but not younger children, report being less satisfied following self-advantaging compared with equal distributions of resources after sharing with another child. One reason that children below grade four might not have reported different emotions to self-advantaging compared equal outcomes in this study is that recipients were absent and anonymous peers. In support of this view, young children's fairness-related behaviour increases when recipients are present compared with when they are absent [31].

In the—to our knowledge—only study, which examined children's spontaneous facial expressions of emotion in response to (un)fairness in a peer context, LoBue *et al.* [32] found that 3- to 5-year-old children respond with negative emotions to receiving less of a resource than a peer, yet with neutral to positive emotions to receiving more. Relatedly, although both younger and older children are aware of the impermissibility of violations of social norms, younger children, between age 4 and 6, have been found to attribute positive emotions to third parties and themselves in hypothetical scenarios

involving transgressions against social norms (e.g. when stealing candy from another child), thus behaving as 'happy victimizers' [33–36]. Only older children, by age 7–9, more consistently attribute negative emotions to transgressors and themselves in such scenarios (e.g. [33]). The overall conclusion from this line of work is that young children are aware of fairness norms from an impartial third-party point of view and respond with negative emotions to disadvantageous inequity. Yet, young children may not experience fairness norms as personally binding to the degree that they express negative emotions when a resource distribution advantages them.

A perhaps critical methodological difference between the study by LoBue *et al*. [32] and prior work on children's social emotions is that children in this study did not cause the negative outcome (i.e. the unfairness) themselves, and rather merely witnessed an adult causing it. Yet, viewing the self, rather than another, as the source of a negative or positive outcome is a central antecedent of social emotions such as guilt, shame and pride [37–39]. For instance, 3-year-olds are more likely to show guilt-like responses when they cause harm to another compared with when no harm is caused or the harm is caused by a third party [39]. Moreover, by age 4–6, preschoolers attribute negative, rather than positive, emotions to transgressors who *unintentionally* violate social norms [35]. Consistent with this prior work on children's social emotions, negative emotions to advantageous inequity may be stronger when children unintentionally cause, rather than passively witness, an advantageous resource allocation. In addition, despite extensive work documenting that adults and children often express social emotions through the body—with an elevated body posture signalling positive social emotions, and a lowered body posture indicating negative social emotions ([40–45]; see [46] for a review)—no previous work has examined whether children express social emotions in response to (un)fairness through their body posture.

Here, we assessed children's emotions, expressed through their body posture elevation, following equal (1,1) and unequal (4,1; advantageous) distributions of resources (within-subjects factor). We tested children between the age of 4 and 10 years (based on [8,18]). In a social context, children divided resources between themselves and a same-aged peer, rendering the resource distributions either fair (1,1) or unfair in favour of the child in focus (4,1). In a non-social control context, children divided resources between themselves and a container 'for resources that no one will receive at the end of the game' rendering the resource distributions merely equal (1,1) or unequal (4,1).

The general design of the current study involved the child in focus (the actor) dividing resource boxes which contain different quantities of a resource (either four stickers or one sticker). Prior to deciding how to divide these resource boxes between themselves and their peer (social context) or a container (non-social context), actors are ignorant with regards to the contents of the boxes. After the actor has decided how to divide the resource boxes, children find out whether the actors' choice created an equal or unequal split (see also [47,48] for similar manipulations). Choosing boxes from a position of ignorance with regards to their content, might lead adults to perceive the outcome as the result of luck, which may lead to a blunted emotional response. However, it appears that having some form of responsibility for a negative outcome, even if it is illusory (e.g. if the outcome is decided on by rolling a die), can lead to negative emotions in children [48]. Shortly following the equal and unequal resource distributions, children's body posture is measured by means of a depth sensor imaging camera (*Kinect*, ©Microsoft) which allows for an objective examination of the change in children's emotions [49,50].

## 1.1. Hypotheses

In sum, we systematically examined if children show a more negative social emotion in response to advantageous inequity than in response to equal distributions of resources expressed through a change in their body posture elevation. Regarding the development of children's emotions, we had two broad hypotheses. If only older children (by age 7–9). respond with negative social emotions to advantageous inequity, children by this age—but not younger children—should show more negative emotions to (4,1) than to (1,1) distributions in the social context (*H1: Emotion and behaviour co-emerge*). This hypothesis is supported by a large body of work showing that children by this age, but not before, reject advantageous resource allocations (e.g. [14,18]). Alternatively, if younger and older children show similar social emotions in response to advantageous inequity, then children across the entire age range should respond with more negative emotions to (4,1) than to (1,1) distributions in the social context (*H2: Emotion precedes behaviour*).

Next, we highlight more specific potential patterns of data that could emerge from the current study (see also electronic supplementary material, appendix S1, section S3). First, in the non-social context

children are predicted to respond with positive emotions to gaining a greater share of a resource on (4,1) trials and more negative emotions in response to receiving a smaller share of a resource on (1,1) trials [47,48]. Consistent with the literature on the development of relief and regret, there may be age-related changes in children's emotional responses to receiving more or less than they could have in the non-social context (see [51] for a review). It is currently unclear when precisely children begin to feel emotions such as relief and regret, as some studies suggest that regret emerges as early as age 4–5 [48], while other studies find that children do not show regret until they are age 6–7 [47]. Moreover, in most studies, relief has emerged later during development than regret [51]. Yet these studies converge on the finding that, if there is a developmental change in the emotions relief and regret between age 4 and 9, then children's positive emotions to receiving more and negative emotions to receiving less than they could have in non-social contexts ought to increase with age, as they become more similar to the adult versions of these emotions.

In the social context, *H1* and *H2* make different predictions only regarding the emotional responses of younger children. Both hypotheses predict that older children by age 7–9, will respond with a more negative emotion to disadvantaging a peer than to equal distribution of resources. For children between age 4 and 6, two data patterns are consistent with *H1: Emotion and behaviour co-emerge*. First, consistent with the findings of Kogut [30], younger children may respond with a similar emotion to unequal and equal distributions of resources or, alternatively, consistent with the findings of LoBue *et al.* [32], younger children might respond with more positive emotions to gaining more of a resource on (4,1) trials compared with (1,1) trials in the social context. By contrast, *H2: Emotion precedes behaviour* predicts that already young children, by age 4–6, respond with more negative emotions to advantageous inequity than to equal resource distributions.

Following the emotion-eliciting phase, children participated in a short behavioural paradigm to assess their willingness to pay a cost to avoid advantageous inequity. Like in a large body of previous work (e.g. [14,15,18,22]), here, we expected only older children (by age 7–9) to reject advantageous resource distribution in the social context (*H3: Late emergence of behaviour*). The reason for including this measure is to provide an outcome-neutral control that the value, ratio and resource chosen can elicit a rejection of advantageous inequity, an established measure for inequity aversion, in the current sample of children.

# 2. General method

We measured children's emotions by means of a depth sensor imaging camera (*Kinect*). This camera, operated through a script run in Matlab (v. 9.5), records the $x$-, $y$- and $z$-coordinates of 20 body posture joints at regular intervals during recorded sequences. In subsequent processing steps, sequences during which children are walking towards the camera are selected based on pre-processing steps developed by Hepach *et al.* [50]. During the study, three baseline recordings, as well as eight test recordings (four following equal and four following unequal resource distributions) were created of the 'actors' (i.e. the child in focus, who divided the resources) while they walked towards the *Kinect* camera. The analyses focused on children's baseline-corrected chest height (the $y$-value of the chest centre data point) and baseline-corrected hip height (the $y$-value of the hip centre data point) during the test recordings [49,50]. In addition, an exploratory analysis examined the change in children's chest expansion (the change in chest height minus the change in hip height; see [52]). Since children did not provide an equal number of data points for each recording, missing values were filled using an interpolation algorithm.

The *Kinect* system, as well as the pre-processing steps designed to extract valid sequences, provide objective measures of relatively small changes (e.g. as small as 1 cm) in postural elevation. For instance, Hepach *et al.* [49] showed that children's posture is lowered when they fail to achieve a positive outcome for themselves, and more elevated when they receive a fun reward to play a game. Similarly, this validation study showed that adults' posture is more slumped when they imagine negative emotions compared with positive ones. More importantly, recent work using this method has shown that 5-year-old children's upper body posture is slumped in response to not helping others [52], providing proof of concept that the methodological approach is able to show children's postural decrease in prototypical guilt- or shame-eliciting situation. While the *Kinect* camera provides a useful tool to examine changes in children's emotions ranging from more positive to more negative, it is important to note that the body posture outcome measures do not objectively differentiate specific

**Table 1.** Model structure of the statistical models for the body posture analysis.

| model | fixed effect | hypothesis |
|---|---|---|
| 1a | context × distribution × age | H1: Emotion and behaviour co-emerge |
| 1b | context × distribution × age × time-distance | emotion and behaviour co-emerge, children's postural change varies across distance |
| 2a | context × distribution | H2: Emotion precedes behaviour |
| 2b | context × distribution × time-distance | emotion precedes behaviour, children's postural change varies across distance |

emotions (e.g. disappointment from shame), which may partially be accomplished by considering the context in which an emotion was expressed.

# 3. Method

The Stage 1 manuscript received in-principle acceptance (IPA) on 20 December 2019. The approved manuscript, unchanged from the point of IPA, can be found at https://osf.io/xem8k/. The data and analysis scripts supporting the findings of the current study can be accessed via the same link: https://osf.io/xem8k/.

## 3.1. Pilot study and power analysis

We conducted a pilot study to ensure that the procedure was adequate for children within the anticipated age range (see also electronic supplementary material, appendix S1, section S3). We piloted a version with a pseudo-randomized trial order, in which two equal or unequal trials were followed by the rest of the trials randomly, before deciding on a blocked trial order. Piloting was completed as soon as all procedural concerns were addressed—most importantly as soon as children understood the task and could follow the experimenters' instructions.

All models for the analysis of children's change in body posture were calculated with pilot data (table 1 for an overview; $N = 21$ children provided body posture data). As an index of effect size, we report $\Delta R_m^2$ resulting from the difference in model fit between full models including all fixed and random effects, and null models including only the control variables and random effects without the fixed effects of interest. $R^2$ was calculated with the in R-package *MuMin* (multi-model inference, [53]), and using the method described by Nakagawa & Schielzeth [54]. These model comparisons yielded an average effect size of $\Delta R_m^2 = 0.065$ for the inclusion of the fixed effects of interest in Models 1a and 1b and an average effect size $\Delta R_m^2 = 0.015$ for the inclusion of the fixed effects of interest in Models 2a and 2b. Power analyses based on model comparisons were conducted with the R-package *simr* (v. 1.0.3, [55]), and indicated an average power of $1 - \beta = 1$, 95% CI [0.9963, 1] to detect effects of this magnitude with our anticipated sample size ($N = 192$).

Based on our pilot data ($N = 19$ children provided behavioural data) and Study 2 of McAuliffe et al. [8], we assumed an effect size of OR = 5.77 (10% of rejections in younger children and 40% in older children in the social context) for the development of children's rejection of advantageous inequity. With the software G*Power [56], $1 - \beta = 0.99$ power was achieved to detect an effect of this size for the analysis of children's behaviour with the specified sample size.

## 3.2. Ethical approval and COVID-19 measures

This study was approved by the ethics committee of the medical faculty of Leipzig University (IRB number: 169/17). Only children whose parents signed a consent form for their child's participation were included in the sample.

Due to pandemic-related circumstances, E1 and E2 wore a face covering for all of data collection that took place after March 2020. Children from age 6 upwards were advised to wear a face covering as well.

## 3.3. Participants

Participants between 4 and 10 years of age ($N = 192$, $M = 84$ months, range = 48–120 months, 101 girls) were recruited from after-school programmes and preschools in the Leipzig area or invited to participate at the child development laboratory at Leipzig University.

Note that for the hypotheses outlined in our Stage 1 manuscript, we referred broadly to 'younger' (4- to 6-year-old) and 'older' (7- to 9-year-old) children, yet we used children's exact age for all analyses, as preregistered. In addition, to ensure approximately equal sampling across the entire age range, and for descriptive statistics, we split children into three age groups (4- to 5-year-olds, 6- to 7-year-olds, 8- to 10-year-olds; based on similar age groups of [8,14]). Our sample included one 10-year-old (who fell only slightly outside of the anticipated age range of 4 years, 0 months to 9 years, 11 months; this was discovered after the child was recruited to participate). Therefore, we refer to the oldest age group as '8- to 10-year-olds', although all other children in this age group were 8 or 9. Our goal was to recruit approximately $N = 32$ children per age group and context. The anticipated number of children per age group served only to ensure that we sampled children approximately equally across the entire age range, yet the final number of children was not identical in each age group and in each context (see electronic supplementary material, appendix S2, table S1 for a breakdown of how many children participated in each age group and of each gender, and electronic supplementary material, appendix S2, table S2 for details regarding children's age within each context).

In the social context, actors were paired in dyads with a gender-matched passive recipient within ±1.5 years age difference ($N = 95$, $M = 83.5$ months, range = 48–119 months). Children were randomly assigned (via a coinflip) to their respective role of actor or recipient within the dyad. We did not use any pseudo-randomization to fill an age bracket. Gender and age were the preregistered criteria for assigning children to dyads. In addition, we avoided having children participate with their siblings. This issue did not come up while piloting the study. Thus, in all there were three criteria for assigning children to dyads: gender, age and non-sibling status.

Due to pandemic-related restrictions during the time of data collection (February 2020 to December 2021), children were either invited to participate in the laboratory ($n = 70$) or tested in their after-school programme ($n = 55$) or preschools ($n = 67$). This deviated from our preregistered sampling plan to collect all data in schools and preschools. Data collection took place during three phases: T1 ran from February to March 2020 ($n = 38$), T2 from July to October 2020 ($n = 35$) and T3 from June to December 2021 ($n = 119$). No data were collected from March to June 2020, as well as from November 2020 to May 2021 (see electronic supplementary material, appendix S2, table S3 for a breakdown of how many children participated in the laboratory and how many participated in schools or preschools depending on age and context during these time periods).

Based on our preregistered criteria (see Stage 1 protocol), data from six children were excluded from the final analyses. Data were excluded from the analyses, because one child in the dyad became upset during the study ($n = 1$) or because children had to leave their after-school programme ($n = 2$). In both cases, the study was stopped early, and children did not complete all body posture trials and the behavioural measure of advantageous inequity aversion (criterion a). For additional children the body posture pre-processing steps did not yield sufficient body posture data on more than two trials per within-subjects condition ($n = 2$) or there was an equipment error that led to insufficient usable body posture data ($n = 1$; criterion b). As preregistered, data from children who met these criteria were excluded from all further analyses, and children were replaced until the sample size of $N = 192$ was reached.

Further individual trials were excluded, if the child walked backwards ($n = 3$) or crawled on the floor ($n = 3$). In both cases, no upright, forward-moving skeleton could be mapped. In addition, as preregistered, individual trials were excluded if there was a wrong distribution in one of the resource boxes (e.g. [5,1], instead of [4,1] or [1,0], instead of [1,1]; $n = 20$).

## 3.4. Materials and design

This study was based on a mixed design with the between-subjects factor context (social or non-social) and the within-subjects factor distribution (equal or unequal). Equal distributions created a (1,1) split and unequal ones created a (4,1) split, which always favoured the participant child in focus. The final choice of these resource allocations was the result of piloting the procedure with children across a wide age range.

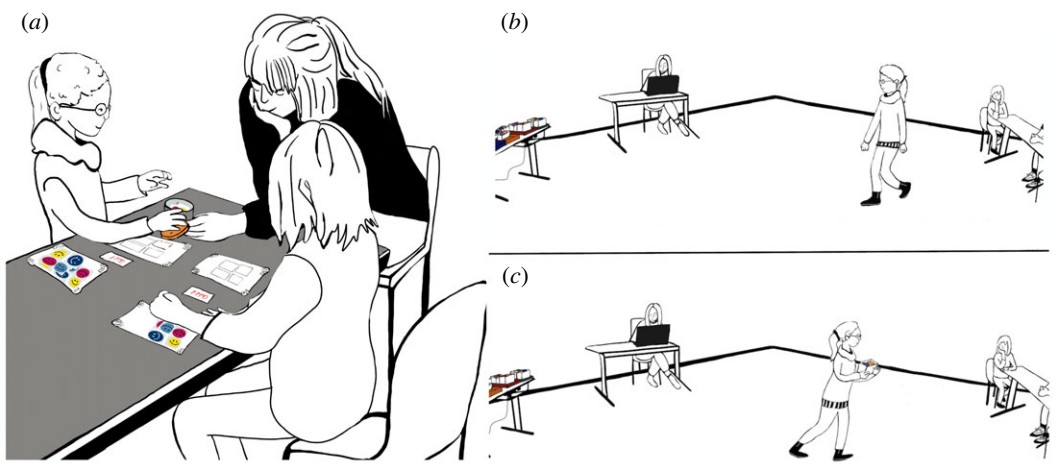

**Figure 1.** Depiction of the study set-up. (a) In the social context, the actor and recipient sat facing each other at the study table. On the study table, each child had a game plate with four squares on it and a collection plate, as well as a name sign. In the non-social context, instead of the recipient, a transparent container was placed on the side opposite the actor. (b,c) The distribution boxes (and resource boxes) were placed on a resource table on the far side of the room. E2 operated the *Kinect* from a laptop in one corner of the room. (b) The *Kinect* camera recorded the actors' body posture while retrieving distribution boxes from the resource table during the baseline phase and after each test trial (only while walking towards the camera and not while walking back). (c) The resource boxes that are retrieved by the actor are used for the following trial. Note that during T2 and T3, E1 and E2 wore facial masks.

During the emotion-eliciting part, children first participated in an introductory phase (two demonstration trials and one practice trial), followed by a test phase (eight test trials, four with an equal and four with an unequal distribution). The order of the equal and unequal test trials was blocked (each child received four trials of the same test trial type in succession followed by four of the other test trial type) and counterbalanced across participants. The blocked design was a result of piloting the procedure, as it appeared most suitable to isolate children's emotions in response to the within-subjects conditions. The dependent measure was children's change in upper body posture following the experimental manipulations. During a post-test phase, a behavioural measure of advantageous inequity aversion was collected [14,22]. Children received one trial with a proposed (4,1) distribution, and choices were coded as 1 (reject; leave resources in the boxes) or 0 (accept; distribute). In addition, as part of efforts to further validate the *Kinect* method children participated in a short post-test interview. Children's responses to the post-test interview questions were not used to directly address the hypotheses and are reported in a separate section on exploratory analyses.

The resources were equally sized colourful smiley stickers [18,32]. Each trial featured a set of resource boxes placed into larger square distribution boxes creating distribution packages of two resource boxes each. The resource boxes, which contained the stickers for each trial, had two compartments, i.e. a double-bottom, allowing the experimenter, E1, to rig the distribution by flipping the resource boxes vertically (figure 2). The distribution boxes were positioned on a resource table and retrieved by the actor during the study (figure 1b). In total, actors retrieved 12 distribution boxes from the resource table: three for the introductory trials, eight for the test trials and one dummy distribution box, which was retrieved following the last test trial, so that actors could provide a final test body posture recording. The distribution boxes for the test trials were placed on coloured trays, so that actors could be instructed regarding which distribution box to pick for each test trial. Children interacted with each other and E1 at a separate study table which was approximately 4.5 m from the resource table. On the study table there were two game plates, on which the resource distributions were placed on each trial, as well as collection plates for children to collect their resources after each trial (figure 1a). In the non-social context, children saw a transparent container with additional resources in it. Note that we referred to the container as 'opaque' in our Stage 1 protocol. This was a mistake, which we have corrected throughout our manuscript. For the behavioural measure of advantageous inequity aversion, resources were placed into a set of separate resource boxes (see also electronic supplementary material, appendix S1, section S1). In the social context, there were also two additional chairs with cardboard screens attached to them next to the two chairs for the actor and recipient (see electronic supplementary material, appendix S1, figure S2). After a coin flip has decided which child would be the actor, the chair in front of the actor was removed.

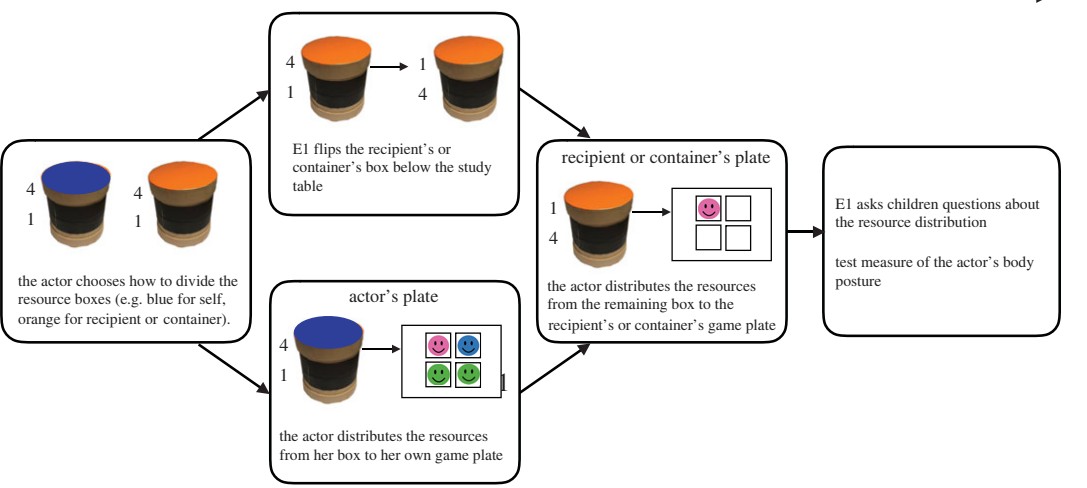

**Figure 2.** A schematic depiction of the sequence of events during each test trial (an unequal trial is shown). Initially, two resource boxes contained the same number of resources in both the top (4) and bottom compartments (1). After the actor has chosen a resource box for each child or herself and the container, the actor opened her own resource box and distributed the stickers from the top compartment to her game plate. Simultaneously, E1 flipped the recipient's or container's resource box, so that instead of four stickers the top compartment of the recipient's or container's resource box now contains one sticker. The actor opened the recipient's or container's resource box and distributed the stickers from the top compartment to the recipient's or container's game plate.

The other chair remained in front of the recipient for the remainder of the study to ensure that only the actor's body posture was recorded by the *Kinect* camera.

## 3.5. Procedure

The study was conducted by two experimenters, E1, who interacted with the children, and E2, who operated the *Kinect* camera from a laptop. Testing took place in a quiet room in children's kindergarten or school or in the child development laboratory.

### 3.5.1. Introductory phase

In the social context, children participated in dyads, and were told that they would be able to collect stickers, which were placed on a table on the far side of the room (see also electronic supplementary material appendix S1, section S2 for a detailed study script). Children were further informed that they would be asked questions while the resources were distributed, and that they were allowed to answer these questions, although it was otherwise a 'quiet game'. Instructing children not to talk during the study was intended to ensure that any potential effects of the disadvantaged recipient's reactions to the actor's resource distributions were comparable across dyads. Following this, a coin flip determined which child became the actor. The other child then became the recipient. In the non-social context, the single child always had the role of the actor. In the social context, E1 briefly removed the chair placed in front of the actor, while the second chair blocked the recipient from the point of view of the Kinect. Then, the actor was asked to retrieve three distribution boxes in succession from the resource table. These contained the resource distributions for the introductory trials. While the actor walked towards the resource table, three baseline recordings of the actor's body posture were taken. Each distribution box for the introductory trials had a different symbol on it. Children were shown each symbol and asked to retrieve the distribution box with the respective symbol on it before walking towards the resource table. These symbols enabled E1 to identify the distribution boxes which contain the correct distribution for the respective introductory trial.

In the social context, E1 explained that each child would get the stickers out of one resource box, and that some contained four stickers while others contained one. Further, children were told that, in the game, they could first place their stickers on their game plates with one sticker occupying a single square (figure 1a), and then move them to their collection plates, where they could collect stickers to take home at the end of the game. E1 wrote each child's name below their respective collection plate,

as a visual reminder of which one belonged to them (or, if children were already able to write, they wrote their name themselves). In the non-social context, the instructions were identical, except that a transparent container with more stickers was placed in the position of the recipient's collection plate. Children were told that no one would get the resources that are returned to the container. After the start of data collection, it became apparent that children frequently gave incorrect answers to a comprehension check regarding what would happen with the resources that were returned to the container. To ensure that the E1's text was clear, the instructions regarding the container were repeated three times for all data collection that took place during T2 and T3.

Following this brief introduction to the game, children received three introductory trials, consisting of two demonstrations of the resource distribution by E1, one with a (4,1; advantageous) distribution, and one with a (1,4; disadvantageous) distribution (order counterbalanced across participants). In addition, the actor received one practice trial, on which she decided how the resource boxes should be divided. For the practice trial the distribution was equal (1,1).

During the practice trial and all subsequent test trials, the actor was asked how the resource boxes should be divided, thus making actors responsible for the outcome (see [48] for the effect of such manipulations on children's emotions in non-social contexts). Thus, in the social context, children were asked: 'Which one should be for (recipient name), the *yellow/orange* or *green/blue* box?', and in the non-social context: 'Which one should go back into the container, the *yellow/orange* or *green/blue* box?' Following the actor's choice of resource box for the recipient or the container, actors were allowed to open the resource boxes and distribute their contents to their own and the recipient's (social context) or the container's (non-social context) game plate.

After each resource distribution, children were asked further questions. In the social context, children were asked to indicate how many stickers they got, whether one child got more, and if so, who did. These questions were designed to prompt children to attend both to their outcome and that of the other child and evaluate the (un)fairness of the resource distribution. In the non-social context children were asked analogous questions, i.e. how many stickers they got, how many will be returned to the container and whether there are more stickers on either the child's game plate or the one with stickers that will be returned to the container. Piloting indicated that children nearly always answered these questions correctly; however, if children answered any question incorrectly, the questions were repeated, and, if necessary, children were corrected. Additionally, in the non-social context, children were asked a comprehension question to ensure their understanding of the container: 'What will happen with the stickers that are in the container at the end of the game?' If children answered this question incorrectly, they were corrected (see also [8]). Fifty-seven per cent of all children in the non-social context spontaneously answered the comprehension check correctly. This proportion increased with age. At age 4–5, 47% of children spontaneously passed the comprehension check, whereas 58% of 6- to 7-year-olds and 69% of 8- to 10-year-olds spontaneously passed the comprehension check.

### 3.5.2. Test phase

After each test trial, the actor was instructed to retrieve a new distribution box from one of the coloured trays (from left to right on the resource table, figure 1b). They were then asked to pick a resource box for each child (social context) or for themselves and the container (nonsocial context) and distribute the resources to the children's or the container's respective game plates. Crucially, the distribution boxes on unequal (4,1) test trials were manipulated, so that the top compartments of the resource boxes each contained four stickers, and the bottom compartments each contained one sticker. Following the actor's choice regarding how the resource boxes should be divided (by selecting one coloured box for each child or for themselves and the container), E1 surreptitiously manipulated the recipient's or the container's resource box by briefly removing it from the table and flipping it vertically below the study table, and outside of children's view. The result of this manipulation was that, on unequal trials, the top compartment of the recipient's or the container's resource box always contained one sticker instead of four, while the top compartment of the actor's resource box still contained four stickers (figure 2). The resource boxes for the recipient or container were flipped on both equal and unequal test trials. On equal test trials all compartments of the resource boxes contained one sticker. During piloting, it was determined that children did not note that E1 rigged the distribution, as no child commented on E1's brief removal of the resource box for the recipient or container.

Shortly following each test trial, the actor was asked to walk towards the resource table to retrieve the distribution box for the next test trial. During this period, test recordings of the actor's body posture were taken using the Kinect (figure 1b). The study continued in the same manner for the next seven test trials.

After the last test trial, when there were no more test distribution boxes left on the resource table, the actor was asked to retrieve a dummy distribution box to provide the last test body posture recording. This distribution box contained an equal (1,1) distribution.

### 3.5.3. Post-test phase

Next, a behavioural measure of actors' advantageous inequity aversion was collected. The actor was shown two new resource boxes and was told that one box (the yellow one) contained four stickers for themselves and that the other box (the green one) contained one sticker for the recipient (social context) or that would be returned to the container (non-social context). The lids of the boxes were lifted briefly to allow children to see how many resources were in each box. The actor was asked if E1 should (1: distribute them/the stickers; coded as 'accept') or should (2: leave them/the stickers in the boxes; coded as 'reject'). The order of answer options 1 and 2 was counterbalanced across participants. This behavioural measure of advantageous inequity aversion parallels inequity aversion paradigms described in Blake & McAuliffe [14] and Shaw & Olson [22], which show that children reject advantageously unequal resource distributions in social settings by age 7–9.

In addition, a post-test interview was conducted, in which children were asked for their self-attributed emotions in response to each outcome. Another unequal (4,1) and equal (1,1) distribution was created (the order was identical to the block order during the test phase) and the actor was asked for each one 'How did you feel when the distribution was like this?' with the options 'good', 'a little bit good', 'a little bit bad' and 'bad' (see [57]).

For part of the data collection (for $n = 76$ children in the social context), the recipient was asked the same post-test interview questions as the actor. The decision to ask the recipient about his or her emotions in response to the resource distribution was intended to provide an opportunity for children to discuss the (un)fairness of the resource distribution after the study's completion and thereby provide a form of debriefing. Recipients' responses are not reported here and will be used to inform future studies. Following the post-test phase, E1 equalized the resource distribution, so that all children left the study with a similar share of stickers.

## 3.6. Coding and reliability of children's behaviour and self-attributed emotions

A research assistant, who was present during the study and operated the Kinect, noted down children's choice to accept or reject the advantageous resource distribution, as well as children's responses during the post-test interview phase. These responses were then entered and, in case of ambiguity, compared with the video recordings by the first author. This approach deviated from the one we preregistered, as we had planned to have the first author code all responses offline from the videos. This modification was introduced to increase efficiency. A second coder, who was blind to the study hypotheses, coded a randomly selected 25% ($n = 48$) of videos. For children's choice to accept or reject the advantageous resource distribution, reliability between the two coders was perfect, Cohen's kappa, $\kappa = 1$.

Children's self-attributed emotions during the post-test interview were coded based on whether they fell into one of four categories (good, a little bit good, a little bit bad or bad) or were incomprehensible. No child's response was incomprehensible. For children's self-attributed emotions, the initial reliability between the first and second coder was as follows: Cohen's kappa, $\kappa = 0.89$. Discrepancies between the first coder and second coder were resolved by re-examining the videos. Afterwards, reliability between the two coders was perfect.

## 3.7. Statistical analyses

### 3.7.1. Body posture

#### 3.7.1.1. Body posture—preregistered confirmatory analysis

The hypotheses regarding the change in children's body posture were tested using linear mixed models (LMMs). Models were calculated in R Studio ([58], v. 1.3.1093 instead of v. 3.3.3) and with the package *lme4* ([59], v. 1.1–25 instead of v. 1.1–15). The dependent measure was the baseline-corrected $y$-value of children's chest-centre point after the experimental manipulations (their change in chest height, see figure 3). A control analysis was conducted on the baseline-corrected $y$-value of children's hip-centre point after the experimental manipulations (their change in hip height). We had no hypotheses for the

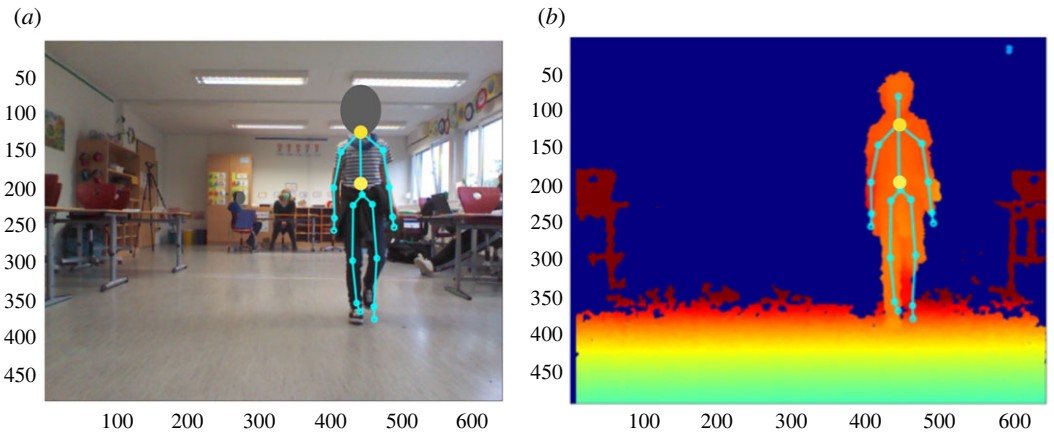

**Figure 3.** An illustration (*a*) of the colour images with skeletal points mapped on them. (*b*) The depth images and skeletal points provided by the *Kinect* camera. Yellow points mark the upper and lower body posture points used for the analyses.

control analysis on children's lower body posture given the stronger evidence for social emotions affecting the upper body [44–46]. However, we considered it possible that the manipulations show a similar effect on the lower body, given that negative social emotions may also elicit a bent-knee gait [60]. The significance of the predictors was determined based on likelihood-ratio tests calculated with the function *drop1()* with the alpha level for statistical significance set at 5%. This analysis strategy parallels previous applications of body posture analyses [50,52].

The preregistered models included random effects for trial, an ID indicating which school or kindergarten children were sampled from or indicating that they came to the laboratory, participant, as well as for children's distance from the *Kinect* camera. In addition, all models included children's gender and trial as control variables (based on [50]). To test *H1: Emotion and behaviour co-emerge*, the interaction of the factors distribution (within-subjects; equal or unequal), context (between-subjects; social or non-social) and age (z-transformed) was included in Model 1a. Based on this hypothesis, we predicted that children's upper body posture would decrease in the social context and on unequal trials to a greater extent for older children. To test *H2: Emotion precedes behaviour*, Model 2a included the interaction of the factors context and distribution. Based on this hypothesis, we predicted that children would show a lower upper body posture in the social context on unequal trials across the entire age range. In addition, Models 1b and 2b examined whether children's change in chest height varied as a function of their distance from the *Kinect* camera (table 1 for an overview). Models 2a and 2b additionally included age as a control variable. The statistical models for the control analysis on children's change in hip height were identical to the above models on children's change in chest height.

We considered it possible that the models would indicate support for both *H1* and *H2*. For instance, both the inclusion of the interaction of context and distribution in Model 2a as well as the inclusion of the interaction of condition, distribution and age in Model 1a might improve model fit. In such a case, if the pattern for each of these interactions was in the predicted direction, our preregistration stated that we would conclude that both *H1* and *H2* have merit in explaining children's change in body posture.

To estimate the age of emergence of a difference between the two within-subjects conditions across social and non-social contexts, we examined at which age 95% confidence intervals of the predicted values for the change in children's body posture were non-overlapping (see also [17] for a similar approach). If more than one of the predictor variables of interest improved model fit in the predicted direction, our preregistered intention was to examine figures with the fitted values for each of the models to determine the age of emergence of a difference between the within-subjects conditions. The same approach was adopted to estimate the age of emergence of a difference in the within-subjects conditions for the change in children's lower body posture, since children's lower body posture also systematically varied as a function of the experimental manipulations.

To summarize, the model structure (for preregistered Model 1a) was as follows:

$$\text{BodyPosture.Change} \ (\text{distribution} + \text{context} + \text{age} + \text{time} - \text{distance})^3$$
$$+ \text{gender} + \text{trial} + (1|\text{School.ID}) + (1 + \text{time} - \text{distance} + \text{trial}||\text{Subject})).$$

### 3.7.1.2. Body posture—exploratory analyses of children's change in chest expansion

We ran additional exploratory analyses with the change in children's chest expansion as a dependent measure. The change in children's chest expansion is the result of subtracting the change in children's hip height from their change in chest height. This measure is informative as to whether patterns seen for children's change in chest height (upper body posture) are explained by children's change in hip height (lower body posture). This could for instance be the case if children's upper body posture varies due to crouching, jumping or running. A recent body of work has shown that children's change in chest expansion is a valid index of the valence of children's emotional response [52,61]. For instance, in one study, the patterns for children's change in chest expansion were similar to those for emotion valence ratings provided by adult coders (Study 1, [52]). In addition, children's expression of happiness (but not shame, sadness, or anger) was found to be correlated with children's change in chest expansion (Study 2, [52]).

### 3.7.1.3. Body posture—exploratory analyses of the first four trials

To assess the potential influence of block order on children's change in body posture, we conducted secondary analyses by running all models including only the data children provided on the first four trials. For these analyses the type of distribution (equal or unequal) varied only between subjects. We outlined our plan to conduct these analyses of the first four trials in our Stage 1 manuscript (as a supplementary analysis), but since we did not plan to use the results of these analyses to directly address our hypotheses, we refer to these analyses as 'exploratory'. The aim of these analyses was to examine whether the (lack of) effects found in our primary confirmatory analyses might be explained by the effect of one type of distribution (e.g. a negative emotional response to the unequal outcome) carrying over into the next (into children's emotional response to the equal outcome).

In addition, we examined whether children's body posture on the first four trials might be affected by trial number. We reasoned that since children were repeatedly presented with the same type of resource allocation across the first four trials, there could be cumulative effects of the same type of resource distribution on children's emotional response.

### 3.7.1.4. Body posture—exploratory analyses of maximum random slopes models

Finally, for an additional exploratory analysis, we ran all models once again with a modified random slopes structure. These models included the maximal number of random slopes justified by our design, and an additional fixed effect of block order ([62]; although see also [63]). We refer to these models as maximum random slopes models (or MRSMs, see electronic supplementary material, appendix S3 for further details). The purpose of these analyses was to examine if the findings of our preregistered analyses remained robust across different analysis approaches. We note that this analysis has a limited interpretation, because, using the same approach as for our preregistered power analysis, which we used to determine sample size, statistical power to detect the effect predicted by Hypothesis 1 in the MRSM (comparable to Model 1a) was $1 - \beta = 0.6$, 95% CI [0.57, 0.63].

### 3.7.1.5. Body posture—summary of analyses

To summarize, we report the results for three body posture measures:

1. The change in children's chest height—the change in the $y$-value of the chest centre data point (preregistered).
2. The change in children's hip height—the change in the $y$-value of the hip centre data point (preregistered).
3. The change in chest expansion—the result of subtracting children's change in hip height from their change in chest height (exploratory).

We ran two analyses for each of these dependent measures: one including data from all trials (for children's change in chest height this was the preregistered confirmatory analysis we used to support one of the stated hypotheses), and one secondary analysis including data from the first four trials (for this analysis distribution [equal or unequal] varied only between subjects; this analysis was exploratory and was not used to directly support our hypotheses). Finally, for an additional exploratory analysis, we ran all models once again with a modified random slopes structure.

### 3.7.2. Behavioural measure of advantageous inequity aversion—preregistered confirmatory analysis

To examine *H3: Late emergence of behaviour*, which predicts that children will begin to show a costly rejection of advantageous inequity at 7–9 years of age [14,22], we ran a general linear model (GLM) with a binomial response term with children's choice to accept or reject the (4,1) offer as the dependent variable. Children's choices will be coded a 1 = accept (distribute the resources) or 0 = reject (leave in the boxes). The main fixed effect of interest was the interaction of the factors age (z-transformed) and context (between-subjects; social or non-social). The significance of the predictors was examined with the *summary()* function with the alpha level for statistical significance set at 5%. We plotted the probability of rejection based on the model-fitted values with associated 95% confidence intervals as a function of age across the two contexts with a fitted line using ordinary least squares. To estimate the age of emergence of children's rejection of advantageous inequity, we preregistered to examine at what age 95% confidence intervals are non-overlapping across contexts.

### 3.7.3. Self-attributed emotions—exploratory analyses

To analyse children's self-attributed emotions, like for the analysis of change in body posture, we ran a LMM using the *lmer()* function of R-package *lme4* [59]. The dependent variable was self-attributed emotion in response to the post-test interview questions coded as 1 = 'felt bad', 2 = 'felt a little bit bad', 3 = 'felt a little bit good' and 4 = 'felt good'. The model included the three-way interaction of the factors distribution (within-subjects: equal or unequal), context (between-subjects: social or non-social) and age (z-transformed), as well as gender as a control predictor. We included a random intercept for subject and a random slope for distribution. Significance was tested based on a likelihood-ratio test calculated with the function *drop1()*.

In addition, to provide further evidence on the question of whether children's change in upper body posture reflects the valence of their emotional expression (see also [50,52]), we ran correlational analyses between children's change in body posture during the emotion-eliciting phase and their self-attributed emotion during the post-test interview phase. A total of 12 correlational analyses were conducted for all three body posture measures (change in chest height, change in hip height and change in chest expansion; each averaged across a maximum of four trials) and separately for each type of distribution (equal or unequal). In addition, all correlations were run once again while controlling for the effect of age.

### 3.7.4. Does children's change in body posture on unequal trials predict their choice to reject the (4,1) advantageous resource distribution?—Exploratory analyses

Next, we examined whether children's change in chest height on unequal trials (averaged across a maximum of four trials), i.e. their negative emotional expression in response to unequal outcomes, predicted their rejection of the unequal resource allocation. We did so to address the question of whether emotion and behaviour in response to advantageous unfairness reflect the same underlying concern in the social context. To this end, we ran two separate GLMs with a binomial response term with children's choice to accept (approx. 0) or reject (approx. 1) the advantageous resource allocation as a dependent measure. Because our main exploratory research question concerned the relation between children's emotional response to the unequal outcome and choice to reject the unequal outcome *in the social context*, we ran separate models for the social and non-social contexts. In addition, we tested the effect of the interaction of age and change in chest height on unequal trials on the choice to reject the (4,1) unequal resource distribution.

We conducted further control analyses with children's change in chest expansion and change in hip height on unequal trials as a predictor of choice to reject the unequal resource distribution (see electronic supplementary material, appendix S4).

### 3.7.5. Does children's self-attributed emotion in response to the unequal outcome predict their choice to reject the (4,1) advantageous resource distribution?—Exploratory analyses

Finally, we examined whether self-attributed emotion predicted children's choice to reject the advantageous outcome. For these analyses, we proceeded in an analogous way as for the above analyses of the relation between change in body posture and choice to reject the (4,1) resource distribution, only that children's self-attributed emotion (between 1 and 4; continuous) on unequal trials was the predictor variable.

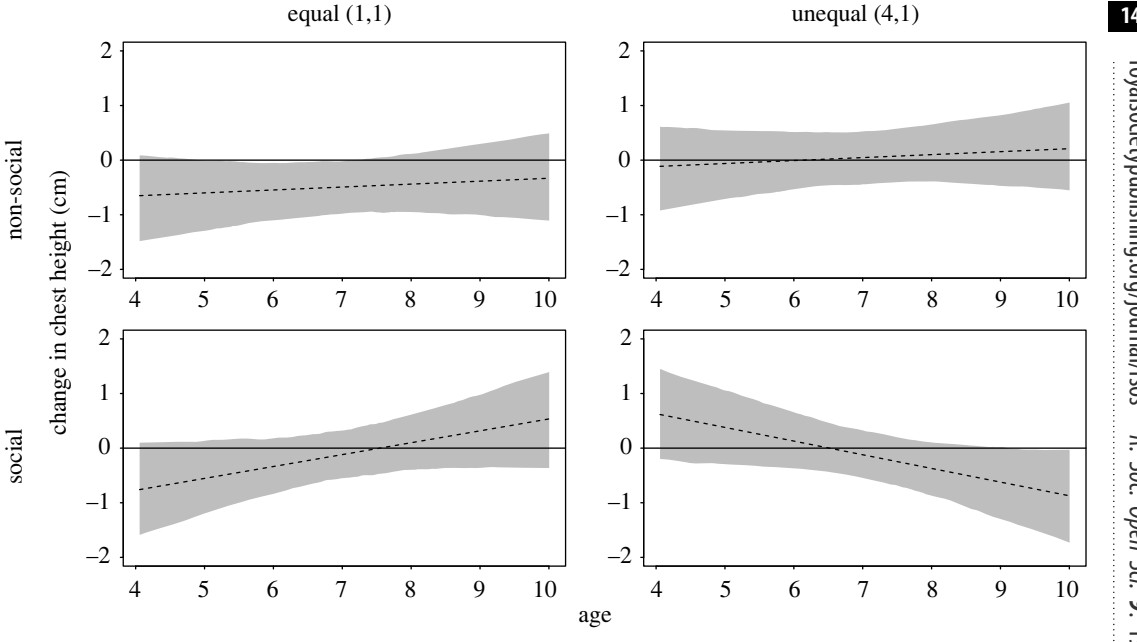

**Figure 4.** Effect of the three-way interaction of context, distribution and age on children's change in chest height based on a linear mixed model (preregistered Model 1a; see electronic supplementary material, appendix S3, figure S2 for a visualization of the raw data). Dashed lines represent the model fitted values; grey areas represent 95% bootstrapped confidence intervals ($N = 1000$), and the solid horizontal line represents the baseline level. This figure shows the fitted values with gender being at its average.

# 4. Results

## 4.1. Body posture

### 4.1.1. Body posture—analyses of all trials

#### 4.1.1.1. Preregistered confirmatory analyses

In our preregistered confirmatory analysis, children's change in chest height varied as a function of the three-way interaction of context, distribution and age (continuous), $\chi_1^2 = 32.44$, $p < 0.001$ (in Model 1a). In the social context, younger children showed greater postural elevation (more positive emotions) when they received more than their peer on (4,1) trials, while their upper body posture was lowered (more negative emotions) on trials where both children received one reward, and the resource distribution was fair (1,1). With an increase in age, the pattern was reversed. Older children in our sample showed a decrease in upper body posture when receiving more rewards than their peer but children's posture was more elevated following fair, equal (1,1) distributions (figure 4). Crucially, we did not find a similar pattern of results in the non-social context, i.e. when no peer was present and only the participating child could receive a reward (see also electronic supplementary material, appendix S3, section S2.1 for details).

Based on visual inspection of figure 4, children until around 5.5 expressed a greater elevation in upper body posture in response to the unequal (4,1) than the equal (1,1) outcome in the social context. By age 8.5 the pattern was reversed. Using our preregistered inference criterion i.e. approximately non-overlapping confidence intervals across unequal and equal resource distributions, disadvantaging a peer on unequal trials of the social context resulted in a greater reduction in upper body posture, a more negative emotional expression, than the equal outcome by approximately age 8.5 (see also electronic supplementary material, appendix S3, table S2 for descriptive statistics).

There was also an interaction of distribution and context that predicted children's change in chest height, $\chi_1^2 = 18.37$, $p < 0.001$ (in Model 2a); however, since the descriptive pattern only aligned with the pattern predicted by Hypothesis 1 and not by Hypothesis 2, we only interpret the three-way interaction, as preregistered.

Further analyses on children's change in hip height revealed a similar pattern to the one observed for children's change in chest height. Specifically, we found that a three-way interaction of context,

distribution and age influenced children's change in hip height, $\chi_1^2 = 9.88$, $p = 0.002$ (see electronic supplementary material, appendix S3, section S2.2).

### 4.1.1.2. Exploratory analyses

Similarly, children's change in chest expansion was predicted by a three-way interaction of context, distribution and age, $\chi_1^2 = 10.05$, $p = 0.002$ (see electronic supplementary material, appendix S3, section S2.3).

We note that aspects of these findings were less robust in exploratory analyses using more maximal random slopes models (MRSMs; see electronic supplementary material, appendix S3, section S2.1 for details). This may influence future studies using different analysis techniques. However, the main pattern of results for children's change in chest height remained descriptively similar when examining the figures based on the fitted values of the MRSMs. The lack of similarly robust effects in the MRSMs thus may have been due to the substantially reduced statistical power in these more complex models compared with the models we preregistered and used to determine sample size.

### 4.1.1.3. Conclusion

Taken together, the results of our preregistered confirmatory analysis suggest that children's entire posture (chest height and hip height) varied as a function of context, distribution and age. In addition, because the findings remained similar in an exploratory analysis with children's change in chest expansion as the dependent variable (i.e. after correcting children's change in chest height for their change in hip height), aspects of the pattern for children's change in chest height were specific to children's upper body posture changing as a function of the experimental manipulations and age.

### 4.1.2. Body posture—exploratory analyses of the first four trials

We found no overall effect of our main predictors of interest (table 1) on children's change in chest height across only the first four trials (see electronic supplementary material, appendix S3, section S2.1), but there was a three-way interaction of context, distribution and trial, $\chi_1^2 = 3.87$, $p = 0.049$. In the social context, children expressed an increasing reduction in their chest height, a more negative emotional expression, after repeatedly disadvantaging a peer on unequal trials. On the other hand, in the social context, repeated equal resource distributions between themselves and others resulted in an increasing elevation in children's chest height, a more positive emotion, across the first four trials (figure 5). In the non-social context, by contrast, children expressed an increasing reduction in their chest height with each equal (1,1) trial, on which children 'lost out' relative to the amount they could have had on unequal trials. There was no additional four-way interaction of context, distribution, age and trial on the first four trials, $\chi_1^2 = 0.1$, $p = 0.752$. Thus, children, independently of age, showed an increasing reduction in their chest height, a more negative emotion, with each trial that disadvantaged a peer.

Children's change in chest expansion also varied as a function of the three-way interaction of context, distribution and trial, $\chi_1^2 = 9.21$, $p = 0.002$, on the first four trials. This suggests that the pattern for children's change in chest height on the first four trials remained similar after correcting children's change in chest height for the change in hip height (see electronic supplementary material, appendix S3, section S2.3). A similar pattern did not emerge for the analysis of children's change in hip height alone, $\chi_1^2 = 0.03$, $p = 0.856$ (see electronic supplementary material, appendix S3, section S2.2), suggesting that this pattern was specific to the change in children's upper body posture.

The findings regarding the effect of context, distribution and trial on the first four trials were furthermore corroborated by the MRSMs (electronic supplementary material, appendix S3, section S2.1), thus, making us confident in the robustness of these patterns across different analysis techniques.

## 4.2. Behavioural measure of advantageous inequity aversion

### 4.2.1. Preregistered confirmatory analysis

There was no strong effect of the two-way interaction of context and age, estimate ± s.e. = 0.64 ± 0.4, 95% CI [−0.13, 1.46], $z = 1.6$, $p = 0.11$, OR = 1.9, on children's choice to reject the (4,1) resource allocation (figure 6). Since the two-way interaction of context and age was not significant, we did not estimate the age of emergence of children's increased rejection of advantageous inequity in the social compared with the non-social setting.

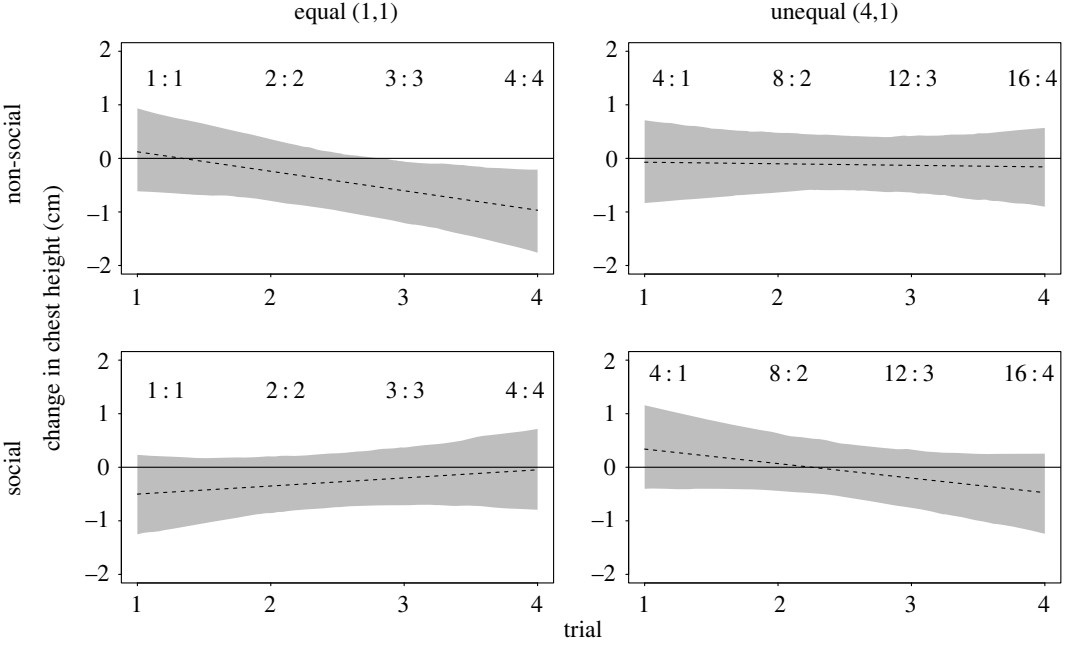

**Figure 5.** Effect of the three-way interaction of context, distribution and trial on children's change in chest height on the first four trials. Fitted lines based on a linear mixed model together with bootstrapped confidence intervals ($N = 1000$). The ratios show the cumulative distribution on the respective trial (e.g. 16 : 4 reads as 16 for self, 4 for other). See electronic supplementary material, appendix S3, figure S6 for plots of the raw data. This plot shows the fitted values with age and gender being at their average.

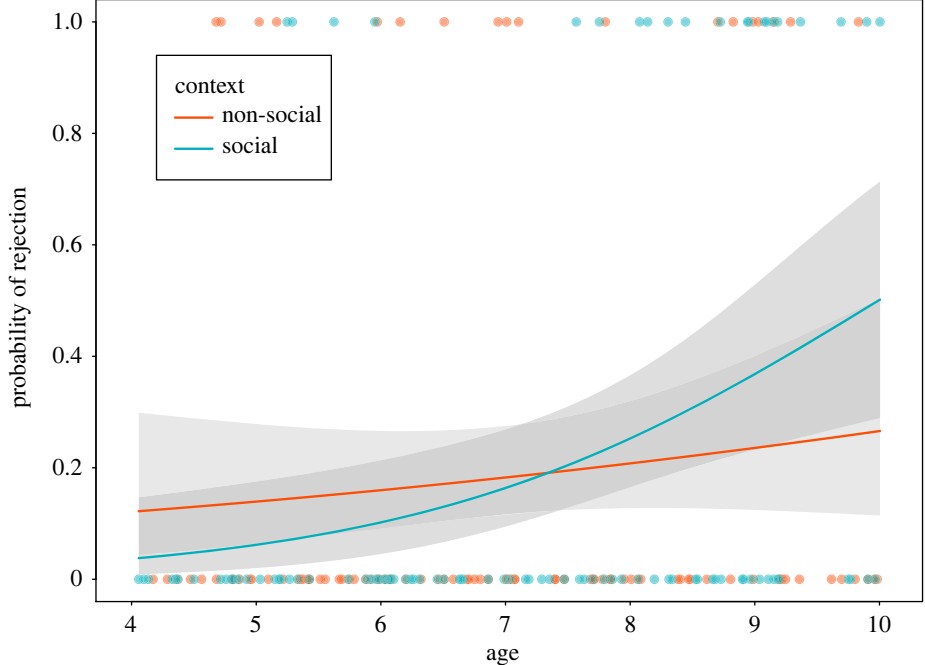

**Figure 6.** Results of a GLM predicting children's rejection of the (4,1) resource distribution based on the two-way interaction of context and age. Grey areas represent 95% confidence intervals. Note that the two-way interaction of context and age did not reach significance, yet we show it in the plot based on our preregistered analysis plan, and because it had a similar effect size to the effect of age. See electronic supplementary material, appendix S4, figure S2 for a plot of the raw data. Dots indicate individual responses for visualization purposes.

### 4.2.2. Exploratory analysis

An additional exploratory analysis was conducted including only the main effects of context, age and gender in the model. This analysis revealed a clear effect of age on children's rejection of the (4,1) unequal resource allocation, estimate ± s.e. = 0.58 ± 0.19, 95% CI [0.21, 0.97], $z = 2.99$, $p = 0.003$, OR =

1.79. With increasing age, children were more likely to reject the advantageous resource allocation independently of context.

### 4.2.3. Descriptive comparison to previous studies

In the present study's social context, 42% of 8- to 10-year-olds rejected the advantageous outcome, compared with 7% of 6- to 7-year-olds and 14% of 4- to 5-year-old children. Older children's rate of rejections of advantageous inequity approximately aligns with the proportion we had predicted in our Stage 1 manuscript (40%; see §3.1 *Pilot study and power analysis*) based on a previous study [8]. At the same time, there was an unexpected age-related increase in children's rejection of the advantageous resource allocation in the non-social setting. In the non-social context, 25% of 8- to 10-year-olds rejected the advantageous outcome compared with 17% of 6- to 7-year-olds and 15% of 4- to 5-year-olds. Previous studies have generally found that children's tendency to reject advantageous outcomes in non-social contexts decreases or remains flat across development [8,19]. We provide a more detailed discussion of the pattern of behavioural results in the non-social context in electronic supplementary material, appendix S4 (section S1).

## 4.3. Self-attributed emotions—exploratory analyses

Children's self-attributed emotions during the post-test interview phase were influenced by a three-way interaction of context, distribution and age (continuous), $\chi_1^2 = 5.72$, $p = 0.017$ (see electronic supplementary material, appendix S4, section S2 and figure S3). Overall, the pattern of results for children's self-attributed emotions mirrors the pattern of results of the preregistered confirmatory analysis of children's change in chest height. Children, with age, reported feeling more negative emotions in response to the unequal than the equal outcome in the social context.

There was, moreover, a significant correlation between children's change in chest height on unequal trials (averaged across a maximum of four trials) and their self-attributed emotion in response to the unequal outcome, Pearson's $r = 0.19$, $p = 0.008$ ($n = 192$). This pattern remained robust in a partial correlation after controlling for age, $r = 0.18$, $p = 0.015$. Similarly, there was a marginally significant correlation between change in chest expansion on unequal trials and self-attributed emotion in response to the unequal outcome, $r = 0.13$, $p = 0.075$, which was of a similar magnitude after controlling for age in a partial correlation, $r = 0.12$, $p = 0.09$. None of the other correlational analyses between the body posture measures and children's self-attributed emotion reached significance (range of $r$s: 0.05–0.11; see also electronic supplementary material, appendix S4, tables S4 and S5). In line with previous studies [50,52], these findings suggest that children's change in chest height and change in chest expansion, but not the change in children's hip height, partly reflect the valence of children's emotional response.

## 4.4. Does children's change in body posture on unequal trials predict their choice to reject the (4,1) advantageous resource distribution?—Exploratory analyses

In the social context, children's change in chest height on unequal trials predicted their choice to reject the advantageous resource allocation, yet this association depended on age, i.e. there was two-way interaction of these factors, $z = -2.08$, $p = 0.038$. To follow up on this effect, we computed the simple slopes for the relation between children's change in chest height on unequal trials and their choice to reject the unequal outcome at three levels of age: 1 s.d. below the mean, at the mean of age, and 1 s.d. above the mean (levels based on the recommendations by Aiken & West [64]) using the R-package *interactions* [65]. These analyses showed that younger children, who expressed a greater *elevation* in upper body posture on unequal trials were *more* likely to reject the advantageous outcome, estimate ± s.e. = 0.57 ± 0.25, $z = 2.32$, $p = 0.02$. This relation was less clear, though went in a similar direction, for children at the mean of age, estimate ± s.e. = 0.25 ± 0.14, $z = 1.71$, $p = 0.09$. For older children in the social context, there was no clear association between children's change in chest height on unequal trials with the choice to reject the advantageous outcome, estimate ± s.e. = −0.08 ± 0.17, $z = -0.45$, $p = 0.65$ (figure 7).

Thus, children, with age, showed a greater tendency to reject advantageous inequity and expressed a more negative emotion (a reduction in their upper body posture) in response to advantageous outcomes in the social context. Nevertheless, older children's change in chest height on unequal trials did not

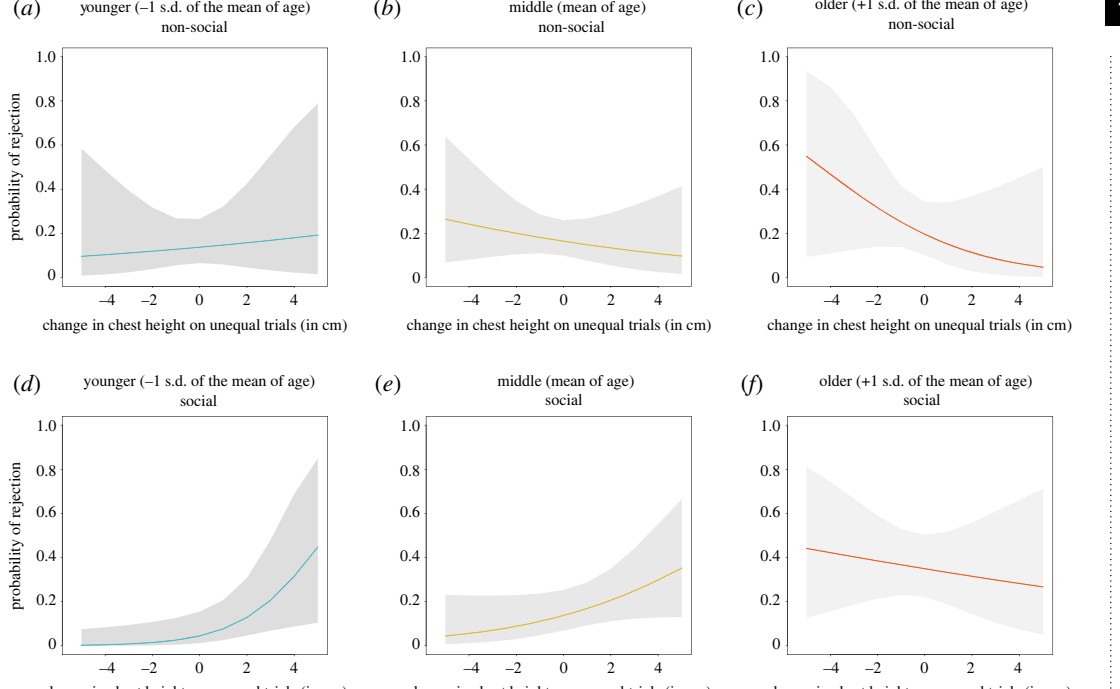

**Figure 7.** Results of GLMs predicting children's choice to reject the unequal outcome in either the social or non-social context from the interaction of the change in chest height and age plotted at three levels of age (1 s.d. below the mean, at the mean of age, and 1 s.d. above the mean of age). Note that the interaction of age and children's change in chest height on unequal trials was only significant in the social context. See electronic supplementary material, appendix S4, figure S7 for a visualization of the raw data. The fitted values for this figure were obtained using the effects package in R [66,67]. In the social context, values were plotted at age 5.4, 7.13 and 8.86, whereas in the non-social context values were plotted at age 5.32, 6.95 and at age 8.59. Four data points fell outside of the prediction interval of −5 to 5 cm of average change in chest height on unequal trials shown here.

predict their choice to reject advantageous inequity in the social setting (figure 7*f*). Younger children who showed a greater *elevation* in upper body posture in response to the unequal outcome, surprisingly, were more likely to reject advantageous inequity in the social setting (see electronic supplementary material, appendix S4, section S3.1 for detailed discussion of this finding).

In the non-social context, there was no two-way interaction of children's change in chest height on unequal trials and age (continuous), $z = -0.98$, $p = 0.327$, nor was there a main effect of children's change in chest height on unequal trials, $z = -0.69$, $p = 0.490$, that predicted children's choice to reject the unequal outcome.

Further analyses with children's change in hip height on unequal trials (averaged across a maximum of four trials) as a predictor variable showed a similar pattern to the results for children's change in chest height (electronic supplementary material, appendix S4, section S3.2). There was no similar pattern of results for the analysis of children's change in chest expansion (electronic supplementary material, appendix S4, section S3.3). Taken together, these analyses suggest that the above finding for children's change in chest height on unequal trials was not specific to the upper body posture predicting children's choice to reject the advantageous resource allocation in the social context.

## 4.5. Does children's self-attributed emotion in response to the unequal outcome predict their choice to reject the (4,1) advantageous resource distribution?—Exploratory analyses

In the social context, children's self-attributed emotion in response to the unequal outcome did not predict children's choice to reject the (4,1) unequal resource distribution, estimate $\pm$ s.e. $= -0.22 \pm 0.28$, $z = -0.81$, $p = 0.415$ (figure 8). Children who reported feeling worse about disadvantaging a peer on unequal trials during the emotion-electing phase were not more likely to reject the advantageous resource distribution during the following behavioural measure of advantageous inequity aversion.

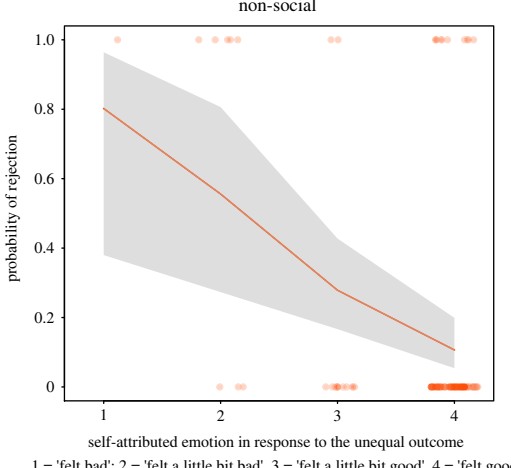
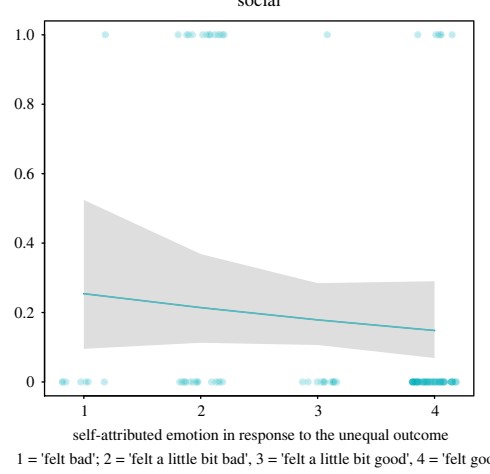

non-social

social

**Figure 8.** Results of GLMs predicting children's choice to reject the unequal (4,1) outcome based on their self-attributed emotion in response to the unequal outcome in the social and non-social contexts. Grey areas show 95% confidence intervals. Dots represent individual data points for visualization purposes. The estimates for this figure were obtained with the effects package in R [66,67].

In the non-social context, on the other hand, children who reported more negative emotions in response to the unequal outcome were also more likely to reject the unequal resource distribution, estimate ± s.e. = −1.18 ± 0.38, $z = -3.08$, $p = 0.002$ (figure 8).

## 5. General discussion

In this study, we investigated 4- to 10-year-old children's emotional responses to (un)fairness measured via changes in children's body posture. Consistent with one of the hypotheses outlined in our Stage 1 manuscript (*H1: Emotion and behaviour co-emerge*), our preregistered confirmatory analysis revealed that children—with age—expressed a greater reduction in their upper body posture following advantageously unfair outcomes and a greater elevation in upper body posture following equal resource distributions between themselves and a peer. In fact, there were two developmental shifts in children's emotional expression in response to advantageous inequity. Until approximately age 5.5, children expressed a more elevated upper body posture (positive emotion) in response to gaining more than a peer on unequal (4,1) trials than in response to the equal (1,1) outcome. On the other hand, children by age 8.5 showed the opposite pattern of results. They expressed a more elevated upper body posture (positive emotion) following the equal (1,1) split than after receiving relatively more than a peer on unequal (4,1) trials. These effects only occurred in a social context in which a peer was disadvantaged compared with the child in focus, and did not emerge in a non-social context, in which children divided resources between themselves and a container. These results align with models of the development of fairness which argue that a concern with equitable—compared with materially advantageous—outcomes is late-emerging and probably requires input from local social and cultural norms (e.g. [17,20,68–70]).

While children's emotional response across all trials was affected by age, there was no clear effect of age on children's emotional response on the first four trials. Over the course of the first four trials, children, independently of age, expressed increasingly negative emotions (lowered upper body posture) in response to disadvantaging a peer in the social context, while equal outcomes between the child in focus and a peer led to a more elevated upper body posture with trial number. The same pattern of body posture changes was not apparent in the non-social context. This suggests that children across the entire age range expressed more negative emotions after disadvantaging a peer with each trial, even while they were getting an increasing number of resources for themselves. Therefore, in these secondary analyses across the first four trials, we find evidence that negative emotional responses to advantageous unfairness emerge by age 4 (which aligns with *H2: Emotion precedes behaviour*), although these emotional responses depend on trial number. The findings regarding the effects of trial on children's change in body posture align with prior studies showing that 3- to 4-year-old children share equally with peers when resources are acquired collaboratively [26–28]. Note, however, that these prior findings on children's early emerging concern with

advantageous inequity are limited to collaborative settings, while children, here, were presented with windfall resource distributions. Moreover, young children from at least age 3 onwards have been found to express guilt and shame in response to harming others [39,71]. Thus, the finding that there is, in part, developmental continuity in our sample's emotional response to advantageous (un)fairness aligns with a body of work showing that aversive responses to harm to others and advantageous unfairness emerge during the preschool years.

The pattern of body posture results (the results of our preregistered confirmatory analysis) support *H1: Emotion and behaviour co-emerge*, yet we did not find an entirely parallel pattern of negative emotional and behavioural responses to advantageous inequity in the social context. Both children's negative emotional response to advantageous inequity and children's rejection of advantageously unequal outcomes increased with age in the social context. Yet, while children's change in upper body posture and self-attributed emotions were affected by context, there was no strong effect of context on children's rejection of advantageous inequity. This suggests that different developmental mechanisms support aversive emotional and behavioural responses to advantageous unfairness. Moreover, if children's emotions and behaviour in response to advantageous inequity resulted from the same underlying concern, we would have expected children's negative emotional response to advantageous inequity to predict their choice to reject the unequal outcome in the social setting. This was not the case. Supporting individual difference analyses suggest that older children expressed negative emotions in response to advantageous inequity in the social context without directly acting on this emotional response by rejecting an advantageous offer at the end of the study. In sum, our findings suggest that negative emotional responses to advantageous unfairness developmentally precede behaviours to reduce unfairness towards others.

The findings of our preregistered confirmatory analyses suggest that young children's self-advantaging behaviour is, in part, underpinned by a lack of negative emotions in response to advantageous inequity, which only emerge in middle childhood. An alternative pattern of body posture results to the one observed would have been that children across the entire age range express more negative emotions in response to advantageous unfairness than to equity across all trials. The apparent lack of concern with advantageous unfairness among younger children in prior studies of children's fairness-related decision-making (e.g. in children's dictator game giving, see [72,73]; and rejection of advantageous unfairness, see [21]) could have been due to competing task demands in these studies, which therefore did not reveal young children's concern with unfairness towards others. Indeed, young children may struggle more than older children to inhibit a pre-potent response to accept large shares of a resource (e.g. [24,25]; although see also [23,74]). Thus, prior findings on children's acceptance of unfairness towards others might have reflected a lack of behavioural control required to align one's behaviour with social norms of fairness (see [23,75]), rather than children's underlying concern with fairness towards others. Our findings speak against this interpretation of young children's behavioural responses to advantageous unfairness. Instead, the present findings suggest that children, with age, develop negative emotional responses to advantageous inequity, which, in part, explain the late emergence of children's costly enforcement of fairness towards others.

The finding that young children (until age 5.5 in the present study) respond with positive emotions (elevated upper body posture) to relative advantage aligns with the findings of LoBue *et al.* [32], although this study contrasted advantageous with disadvantageous inequity, thus leaving it unclear whether positive emotions to advantage or negative emotions to disadvantage were driving the effect. By contrast, in the present study, we directly compared children's concern with advantage over fairness (which resulted in a smaller share of resources for the child in focus), showing that relative advantage is emotionally rewarding to preschool children until age 5.5. This finding aligns with the study of Sheskin *et al.* [16], showing that 5- to 6-year-olds sometimes express 'anti-equality,' by taking a cost to avoid fairness and instead put a peer at a relative disadvantage. On the other hand, this finding contrasts with the prior study of Kogut [30] suggesting that preschool children respond with similar emotions to equity and relative advantage. Our results may differ from those of Kogut [30] because we measured children's emotional response in a face-to-face setting, instead of in response to sharing with absent and anonymous peers.

In our study, children expressed increasingly negative emotions in response to advantageous unfairness over development. This finding aligns with prior work showing that children by age 10 report more positive emotions in response to equity than relative advantage [30]. Our interpretation of this finding is that unintentionally violating fairness norms, and thereby advantaging oneself, elicits social, self-conscious emotions, such as shame or guilt, by around age 8.5. Emotions such as shame or guilt are theorized to have the evolved function of motivating socially valued behaviour, including

cooperation [44,60,76–79]. These self-conscious emotions, moreover, provide a motivational foundation to override one's self-interest when self-interest does not clearly align with social norms, standards or values (e.g. [11]). Thus, children's expression of increasingly negative emotions in response to advantageous inequity with age could reflect children gradually applying social norms of fairness to themselves to the extent that even materially advantageous unfairness causes a negative (shame- or guilt-like) emotional response. Similarly, pride may underlie the developmental increase in children's positive emotions in response to creating equitable outcomes, and thereby following fairness norms. Pride often results from behaving in a socially valued, norm-conform manner [76,80].

A further possibility is that older children's negative emotional response reflects a developmental increase in sympathy with the disadvantaged recipient. Sympathy can be defined as an emotion resulting from an understanding of another's emotion and a concerned response to the distress of a needy other [81,82]. Indeed, guilt and sympathy are thought to be highly related. Yet, while guilt has been argued to depend on a sympathetic response combined with an awareness that one is the cause of harm [83], sympathy may emerge in the absence of any (assumed) causal responsibility for the harm (i.e. when children are innocent bystanders and witness the harm being caused). Similarly, shame is thought to depend on a negative self-evaluation after having oneself violated a social standard or norm [37,60]. Thus, to test whether children's emotional response is more sympathy-like or, rather, reflects a negative self-conscious emotion (such as shame or guilt) future studies should manipulate whether children's actions (choosing one box for self and one for other) were instrumental in creating inequity or not (see [39] for a similar study).

The lack of an entirely parallel developmental pattern between children's negative emotional and behavioural responses to advantageous inequity raises the question to what extent children's emotions play a role in motivating behaviours to enforce fairness towards others (e.g. the rejection of advantageous inequity). Based on the current data, one possibility is that children's negative emotions in response to advantageous unfairness, in part, motivate aversive behavioural responses to advantageous inequity over development, given that both kinds of responses increase with age. However, children's emotional and behavioural responses to advantageous unfairness may still not be entirely aligned by age 8–10, because additional cognitive and motivational factors influence children's choice to reject advantageous inequity. For instance, children may need to be able to anticipate a negative emotional reaction for this emotion to influence their decision-making (e.g. [84]). The ability to anticipate emotions (e.g. regret over choices made for oneself) has been found to develop later than their expression [85,86]. In addition, children's choice to reject advantageous inequity is, in part, strategically motivated by a desire to appear concerned with fairness towards observers—especially peers who would be disadvantaged by inequity ([87,88]; see also [89]). Thus, children's rejection of advantageous unfairness may not exclusively depend on a negative emotional response to outcomes that disadvantage others, but require additional strategic motivations. On the other hand children's negative emotional response to disadvantaging others may not depend on similar strategic concerns (see [52], although see also [90]). To test this possibility, future studies should examine whether children's emotional expression and behaviours in response to advantageous inequity are influenced by comparable strategic motives. This could be accomplished by measuring children's emotional and behavioural response to advantageous inequity while varying whether the peer knows that they have been disadvantaged (see [87]).

# 6. Conclusion

Children expressed negative emotions (reduction in upper body posture) with age in response to disadvantaging a peer in a novel resource-distribution task. This emotional response was not explained by children's emotional expression in a non-social context and showed a different developmental pattern than children's choice to reject outcomes that disadvantaged others. These findings support the notion that children over development acquire social norms for fairness resulting in guilt- or shame-like negative emotions after advantaging themselves and thereby violating fairness norms.

Data accessibility. The data and analysis scripts supporting the findings of the current study can be accessed through the Open Science Framework: https://osf.io/xem8k/. The Stage 1 manuscript is available via the same link.

The data are provided in the electronic supplementary material [91].

Authors' contributions. S.C.G.: conceptualization, data curation, formal analysis, investigation, methodology, project administration, visualization, writing—original draft; K.M.: conceptualization, writing—review and editing; P.R.B.:

conceptualization, writing—review and editing; D.B.M.H.: conceptualization, writing—review and editing; R.H.: conceptualization, funding acquisition, resources, software, supervision, writing—review and editing.

All authors gave final approval for publication and agreed to be held accountable for the work performed therein.

Conflict of interest declaration. We declare we have no competing interests.

Funding. S.C.G. was supported by the doctoral grant 'Doktorandenförderplatz' of Leipzig University and received partial support from a D.A.A.D. German Academic Exchange Foundation scholarship for doctoral students while writing this project. This work received partial support through a DFG (German Research Foundation) grant awarded to R.H. (grant no.: HE 7865/3-1).

Acknowledgements. We thank all the research assistants who helped with recruiting, data collection and coding for this study. We thank Leonore Blume, Alina Kuzima, Jasmin Biber, Johanna Seumel, Anna-Luka Schwanna, Lea Jödicke, Dieu Anh Dong, Antonia Duefeld and Jessica Baars for their assistance. Special thanks to Katja Kirsche for her help with recruiting and to Leonore Blume for creating visualizations of the procedure. We thank the participating schools and kindergartens for their cooperation, and the participating children and their parents. In addition, we thank Theo Toppe, Marlene Försterling and Brian Parkinson for helpful comments on an earlier draft of this manuscript. Special thanks to Elmar Tarajan and Anja Neumann for writing the scripts to run the Kinect and to Kim-Laura Speck for help with the Matlab code.

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
