## [Peer Review File · Royal Society Open Science]

Review History

RSOS-191456.R0 (Original submission)

Review form: Reviewer 1

Is the language acceptable?

Yes

Do you have any ethical concerns with this paper?

No

Have you any concerns about statistical analyses in this paper?

No

Recommendation?

Accept with minor revision

Comments to the Author(s)

Please see attached (Appendix A).

Review form: Reviewer 2

Do you have any ethical concerns with this paper?

No

Recommendation?

Accept with minor revision

Comments to the Author(s)

See attached document for all comments (Appendix B).

Review form: Reviewer 3

Do you have any ethical concerns with this paper?

No

Recommendation?

Accept with minor revision

Comments to the Author(s)

The proposed study aims to investigate 4- to 9-year-old children's emotional responses towards fair vs. unfair events. Children's emotional state is measured implicitly and elegantly via their body posture, with a lower body posture representing a negative social emotion and an elevated body posture expressing a positive social emotion. This is a relatively new measure and has not been used in developmental studies (to my knowledge) before.

I find the manuscript very well written, clear and precise in its language and strong in its description of the methods and results. It is definitely contributing to the field and, if going ahead, will shed more light on the emotional as well as motivational underpinnings of children's understanding (and execution) of fairness.

A few thoughts:

p. 4 - The mentioning of shame comes quite abruptly - a further sentence as to how the authors link shame to advantageous inequity or indeed the acknowledgment of fairness would be helpful.

Similarly, I would like the authors to elaborate more on why they think shame is the main negative feeling in reaction to an advantageous resource allocation. What about guilt or embarrassment?

p.6. - I would like the authors to include more information on alternative emotional expressions that might alter body posture (e.g. disappointment) and how they would distinguish alternative negative (or positive) emotions expressed by a similar body posture.

Generally, I would like to encourage the authors to be very careful when generalising from (observable) behaviour to emotional states. For instance, if a study reports that children reject unequal offers it should not be automatically assumed that this overt behaviour is driven by a

(covert) negative emotion, unless additional evidence has been presented (e.g. coding of facial expressions, verbal responses, etc.).

A few ideas:

Why do the authors not measure the body posture of the recipient as well as the actor to collect some data on recipients' emotional response and to investigate whether there is a link between actors' and recipients' emotional reactions? I understand that this would not be possible for the non-social condition, but it would add an interesting layer of additional data and analysis. The method could be adjusted in a way that recipients are "involved" in the final allocation decision as well (e.g. making it a 2-step process, 1. Step: Recipient blindly chooses 2 resources boxes out of many, 2. Step: Actor blindly decides on their allocation).

Similarly, why do the authors not extend the study to disadvantageous inequity aversion, i.e. the actor receiving less (1;4). In their introduction the authors mention the limited number of studies looking at children's emotional reactions in unequal allocation situations per se. In including a disadvantageous allocation, the authors would be able to support previous findings that these scenarios elicit disappointment.

It might be thoughtful to pilot different resource allocations before final data collection. An allocation of 5:1 or 6:1 might lead to stronger (and therefore presumably more measurable) emotional expressions, especially for the younger age group.

Decision letter (RSOS-191456.R0)

23-Oct-2019

Dear Ms Gerdemann,

On behalf of the Editors, I am pleased to inform you that your Stage 1 Registered Report RSOS-191456 entitled "The ontogeny of children's social emotions in response to (un)fairness" deemed suitable for in-principle acceptance in Royal Society Open Science subject to minor revision in accordance with the referee and editor suggestions. Please find their comments at the end of this email.

The reviewers and handling editors have recommended publication, but also suggest some minor revisions to your manuscript. Therefore, I invite you to respond to the comments and revise your manuscript.

Please submit the revised version of your manuscript within 7 days (i.e. by the author due date is unavailable). If you do not think you will be able to meet this date please let me know immediately.

When submitting your revised manuscript, you will be able to respond to the comments made by the referees and upload a file "Response to Referees" in "Section 6 - File Upload". You can use this

to document any changes you make to the original manuscript. In order to expedite the processing of the revised manuscript, please be as specific as possible in your response to the referees.

Full author guidelines can be found here <https://royalsocietypublishing.org/rsos/registered-reports#ReviewerGuideRegRep>.

Kind regards,
Lianne Parkhouse
Editorial Coordinator
Royal Society Open Science
openscience@royalsociety.org

Associate Editor Comments to Author (Professor Chris Chambers):

Three expert reviewers have now assessed the manuscript. All are generally positive about the proposal but note a range of areas that require clarification and possible revision, including consideration of additional literature underpinning the rationale, clarity of the hypotheses, adequacy of the power analyses and controls, and possible consideration of pilot data. All issues appear to be readily addressable in a thorough revision and response.

Reviewer comments to Author:

Reviewer: 1
Comments to the Author(s)
Please see attached.

Reviewer: 2
Comments to the Author(s)
See attached document for all comments.

Reviewer: 3
Comments to the Author(s)
The proposed study aims to investigate 4- to 9-year-old children's emotional responses towards fair vs. unfair events. Children's emotional state is measured implicitly and elegantly via their body posture, with a lower body posture representing a negative social emotion and an elevated body posture expressing a positive social emotion. This is a relatively new measure and has not been used in developmental studies (to my knowledge) before.

I find the manuscript very well written, clear and precise in its language and strong in its description of the methods and results. It is definitely contributing to the field and, if going ahead, will shed more light on the emotional as well as motivational underpinnings of children's understanding (and execution) of fairness.

A few thoughts:

p. 4 - The mentioning of shame comes quite abruptly - a further sentence as to how the authors link shame to advantageous inequity or indeed the acknowledgment of fairness would be helpful.

Similarly, I would like the authors to elaborate more on why they think shame is the main

negative feeling in reaction to an advantageous resource allocation. What about guilt or embarrassment?

p.6. – I would like the authors to include more information on alternative emotional expressions that might alter body posture (e.g. disappointment) and how they would distinguish alternative negative (or positive) emotions expressed by a similar body posture.

Generally, I would like to encourage the authors to be very careful when generalising from (observable) behaviour to emotional states. For instance, if a study reports that children reject unequal offers it should not be automatically assumed that this overt behaviour is driven by a (covert) negative emotion, unless additional evidence has been presented (e.g. coding of facial expressions, verbal responses, etc.).

A few ideas:

Why do the authors not measure the body posture of the recipient as well as the actor to collect some data on recipients' emotional response and to investigate whether there is a link between actors' and recipients' emotional reactions? I understand that this would not be possible for the non-social condition, but it would add an interesting layer of additional data and analysis. The method could be adjusted in a way that recipients are "involved" in the final allocation decision as well (e.g. making it a 2-step process, 1. Step: Recipient blindly chooses 2 resources boxes out of many, 2. Step: Actor blindly decides on their allocation).

Similarly, why do the authors not extend the study to disadvantageous inequity aversion, i.e. the actor receiving less (1;4). In their introduction the authors mention the limited number of studies looking at children's emotional reactions in unequal allocation situations per se. In including a disadvantageous allocation, the authors would be able to support previous findings that these scenarios elicit disappointment.

It might be thoughtful to pilot different resource allocations before final data collection. An allocation of 5:1 or 6:1 might lead to stronger (and therefore presumably more measurable) emotional expressions, especially for the younger age group.

Author's Response to Decision Letter for (RSOS-191456.R0)

See Appendix C.

RSOS-191456.R1 (Revision)

Review form: Reviewer 1

Do you have any ethical concerns with this paper?

No

Recommendation?

Accept in principle

Comments to the Author(s)

I would like to thank the Authors for their responsiveness to my previous comments. I do not have any further comments at this time.

Review form: Reviewer 2**Do you have any ethical concerns with this paper?**

No

Recommendation?

Accept in principle

Comments to the Author(s)

The authors have adequately addressed all the concerns raised in the reviews.

I am happy with the authors' responses to my requests for revisions. I think the present version is very stringent, and I find the inconsistencies and open issues that were raised in the various reviews are addressed satisfactorily.

Review form: Reviewer 3**Do you have any ethical concerns with this paper?**

No

Recommendation?

Accept in principle

Comments to the Author(s)

I am happy with the responses and amendments that the authors made in relation to my comments. I am happy for this study to go ahead and curious to read about the results. I am wishing the authors the best of luck with their study.

Decision letter (RSOS-191456.R1)

20-Dec-2019

Dear Ms Gerdemann

On behalf of the Editor, I am pleased to inform you that your Stage 1 Registered Report RSOS-191456.R1 entitled "The ontogeny of children's social emotions in response to (un)fairness." has been accepted in principle for publication in Royal Society Open Science. The reviewers' and editors' comments are included at the end of this email.

You may now progress to Stage 2 and complete the study as approved. Before commencing data collection we ask that you:

- 1) Update the journal office as to the anticipated completion date of your study.
- 2) Register your approved protocol on the Open Science Framework (<https://osf.io/rr>) or other recognised repository, either publicly or privately under embargo until submission of the Stage 2 manuscript. Please note that a time-stamped, independent registration of the protocol is mandatory under journal policy, and manuscripts that do not conform to this requirement cannot be considered at Stage 2. The protocol should be registered unchanged from its current approved state, with the time-stamp preceding implementation of the approved study design.

Following completion of your study, we invite you to resubmit your paper for peer review as a Stage 2 Registered Report. Please note that your manuscript can still be rejected for publication at Stage 2 if the Editors consider any of the following conditions to be met:

- The results were unable to test the authors' proposed hypotheses by failing to meet the approved outcome-neutral criteria.
- The authors altered the Introduction, rationale, or hypotheses, as approved in the Stage 1 submission.
- The authors failed to adhere closely to the registered experimental procedures. Please note that any deviations from the approved experimental procedures must be communicated to the editor immediately for approval, and prior to the completion of data collection. Failure to do so can result in revocation of in-principle acceptance and rejection at Stage 2 (see complete guidelines for further information).
- Any post-hoc (unregistered) analyses were either unjustified, insufficiently caveated, or overly dominant in shaping the authors' conclusions.
- The authors' conclusions were not justified given the data obtained.

We encourage you to read the complete guidelines for authors concerning Stage 2 submissions at <https://royalsocietypublishing.org/rsos/registered-reports#ReviewerGuideRegRep>. Please especially note the requirements for data sharing, reporting the URL of the independently registered protocol, and that withdrawing your manuscript will result in publication of a Withdrawn Registration.

Once again, thank you for submitting your manuscript to Royal Society Open Science and we look forward to receiving your Stage 2 submission. If you have any questions at all, please do not hesitate to get in touch. We look forward to hearing from you shortly with the anticipated submission date for your stage two manuscript.

Kind regards,
Anita Kristiansen
Editorial Coordinator
Royal Society Open Science
openscience@royalsociety.org

on behalf of Professor Chris Chambers (Registered Reports Editor, Royal Society Open Science)
openscience@royalsociety.org

Associate Editor Comments to Author (Professor Chris Chambers):

Associate Editor: 1

Comments to the Author:

All reviewers are satisfied with the revised submission and the manuscript can be now be awarded in-principle acceptance.

Reviewers' comments to Author:

Reviewer: 1

Comments to the Author(s)

I would like to thank the Authors for their responsiveness to my previous comments. I do not have any further comments at this time.

Reviewer: 2

Comments to the Author(s)

The authors have adequately addressed all the concerns raised in the reviews.

I am happy with the authors' responses to my requests for revisions. I think the present version is very stringent, and I find the inconsistencies and open issues that were raised in the various reviews are addressed satisfactorily.

Reviewer: 3

Comments to the Author(s)

I am happy with the responses and amendments that the authors made in relation to my comments. I am happy for this study to go ahead and curious to read about the results. I am wishing the authors the best of luck with their study.

Author's Response to Decision Letter for (RSOS-191456.R1)

See Appendix D.

Decision letter (RSOS-191456.R2)

Dear Ms Gerdemann:

I am pleased to inform you that your manuscript entitled "The ontogeny of children's social emotions in response to (un)fairness." is now accepted for publication in Royal Society Open Science.

Please ensure that you send to the editorial office an editable version of your accepted manuscript, and individual files for each figure and table included in your manuscript. You can send these in a zip folder if more convenient. Failure to provide these files may delay the processing of your proof.

Please remember to make any data sets or code libraries 'live' prior to publication, and update any links as needed when you receive a proof to check - for instance, from a private 'for review' URL to a publicly accessible 'for publication' URL. It is also good practice to add data sets, code and other digital materials to your reference list.

Royal Society Open Science is a fully open access journal. A payment may be due before your article is published. Our partner Copyright Clearance Center's RightsLink for Scientific Communications will contact the corresponding author about your open access options from the email domain @copyright.com (if you have any queries regarding fees, please see <https://royalsocietypublishing.org/rsos/charges> or contact authorfees@royalsociety.org).

on behalf of Professor Chris Chambers (Subject Editor).

Follow Royal Society Publishing on Twitter: @RSocPublishing
Follow Royal Society Publishing on Facebook:
<https://www.facebook.com/RoyalSocietyPublishing/>
Read Royal Society Publishing's blog:
<https://royalsociety.org/blog/blogsearchpage/?category=Publishing>

Appendix A

Summary

This pre-registered report has been developed to test whether 4- to 9-year-old children exhibit positive (i.e., elevated posture) or negative (i.e., lowered posture) emotion in response to equal or unequal resource distributions in a social or non-social context. This is an important area of study that will advance current understanding of the link between emotions and children's inequity aversion/resource distribution behaviour. The Introduction, Methods and Proposed Analyses are generally clear and well-written. Below I provide my review in the context of addressing the key points required by reviewers as per the Reviewer Guidelines.

1. The scientific validity of the research question(s).

The research questions examined in the proposed research are important and addressing them will advance our understanding of the link between emotions and perceptions of fairness, a topic that has not been adequately researched in the field despite having significant implications for debates surrounding the role that nature and/or nurture play in human social behaviour. The findings are likely to be of interest to a broad readership.

2. The logic, rationale, and plausibility of the proposed hypotheses.

The hypotheses presented are logical and well-motivated.

However, I thought that the wording of H1 and H2 could be clearer. H1, is over 4 lines of text which makes it difficult to unpack. I did appreciate the Figures of the hypotheses in the supplementary materials which were very helpful depictions of the various hypotheses.

3. The soundness and feasibility of the methodology and analysis pipeline (including statistical power analysis where applicable).

The methodology is appropriate for testing the research questions of interest. The analysis pipeline is based on prior work that has successfully used this measure of body posture as an index of emotion is clearly described. Justification for the sample size was provided with appropriate power analyses using prior work and a pilot sample.

4. Whether the clarity and degree of methodological detail would be sufficient to replicate exactly the proposed experimental procedures and analysis pipeline.

Although the registered report and the supplementary materials provided a good amount of detail regarding the procedure, there were a few points that remained unclear after a few reads of both materials.

Firstly, I remain somewhat unclear as to the meaning of “vertically flipping” and how it will be carried out. A photo or schematic of how this will actually be carried out would be helpful.

The manuscript was also missing details on how the dyads will be determined. Will this be random? I also wondered if the authors could consider attaining teacher ratings of the children in the class and/or how friendly the children within each dyad are to one another. It is my expectation that friendship status is likely to play a role in driving children’s decision making in these sorts of contexts. This might be particularly important when examining differences between the social and non-social conditions.

In the Analysis section page 19, the Authors state that “trial” will be included as a random effects and “trial” as a control. Is it possible for trial to be included as both a fixed effect and a control variable? I’m wondering if one of these is meant to be “order” or “block”?

5. Whether the authors provide a sufficiently clear and detailed description of the methods to prevent undisclosed flexibility in the experimental procedures or analysis pipeline.

The authors clearly describe the processes for excluding participants and/or trials, which I believe will prevent “undisclosed flexibility in the experimental procedures”.

I had just one question, how will the Authors identify trials in which the incorrect distribution was given (i.e., will this be completed post hoc via offline coding?).

6. Whether the authors have considered sufficient outcome-neutral conditions (e.g. absence of floor or ceiling effects; positive controls; other quality checks) for ensuring that the results obtained are able to test the stated hypotheses.

The non-social condition, the blocked design, the behavioural measure and the post-behavioural questions provide the authors with several means to examine potential alternative explanations of the findings and thus, give confidence that the results will provide an appropriate test of the stated hypotheses.

One potential difference between the two conditions that could impact the results is the difference between the likelihood of children in the social condition receiving facial feedback from their partner vs no possibility of facial feedback in the non-social condition. The authors state that children will be told that it is a “quiet” game to reduce the possibility that any verbal information could interfere with the performance which is a strength of the procedure, however, there is still the possibility of children receiving non-verbal feedback. Is there a way that the authors could reduce the possibility of this interfering with/accounting for the results? Perhaps, completing some post-hoc coding of the sessions might be a better option?

I also noted above, that another possible confound source of differences between conditions could be actor child’s preference for the other child in their dyad. Is there a way that the authors could control for this affecting their results?

Other points:

- Page 9, line 45: missing a “.”
- Page 8, line 38: Suggest changing “when” to “if” since the research questions are quite novel.
- In the Abstract, I think it would be clearer if the Authors use the term “actor” to specify the child who will be making the selection (as they do within the manuscript) to minimize opportunities for confusion.
- Regarding the Introduction:
 - The Authors present important terms like “social context” and “social emotions” without providing additional information as to what they mean. For example, on page 4 lines 22-28, the authors raise the findings relating to disadvantageous inequity resulting in anger particularly in “nonsocial contexts”. Given the broad readership of this outlet, I think that the introduction could be strengthened by providing a (brief) section in which the distinction between social and non-social contexts is provided, as well as what the authors mean when they are referring to “social emotions”.
- Page 10, line 19: “:” in the Dunham et al., 2018 reference should be “;”
- Page 17, line 33: I thought this line was confusing and suggest “E1 will write children’s name...” be changed to “E1 will write the children’s name...” or “..each child’s name...”

Appendix B

The suggested study “The ontogeny of children’s social emotions in response to (un)fairness” wants to examine the ontogeny of children’s shame-like social emotion in response to advantageous inequity. Therefore, a paradigm is applied that measures body postures in children between 4-9 years of age. The authors present a sound study with an interesting paradigm to capture social emotions in order to investigate a relevant research question.

Introduction

The assumptions are embedded in a sound theoretical and empirical framework. The authors start the introduction by bringing up prior findings and theories on the development of social emotions and inequity aversion. The strong human concern for fairness and inequity aversion are presented as the base of this research. Relevant findings related to these two aspects in adults and children are presented. However, there is no reference to the happy victimizer paradigm (e.g., Nunner-Winkler & Sodian, 1988; Nunner-Winkler, 1998; Malti & Krettenauer, 2013) which would enhance the literature review. Research from this field also demonstrated that although children know and value specific norms (such as fairness) from early age on, they however do feel good when they experience advantageous unfairness themselves (e.g., Beißert, Mulvey & Killen, 2018) and they also expect others to feel good as well when receiving an unfair reward (e.g., Nunner-Winkler, 1998). That is, children do strive for fairness and equity, however at least young children typically do feel good in the case of advantageous fairness. It might be helpful, to include this field of research in the theoretical considerations and relate it to the current research.

Also, this phenomenon might be relevant for the current hypotheses. Given the happy victimizer phenomenon, it could be expected that children in the age of the planned sample might still be happy victimizers to some extent (that is, they could show positive emotions in the case of advantageous inequity). Research showed that until age 6-9 (or even older), there is still a considerable amount of children who are happy victimizers. In my opinion, the authors’ hypotheses are still valid, though. But it would enhance the rationale to include this aspect in the theoretical considerations.

Methods

The authors give a clear and convincing explanation that the presented paradigm is adequate to analyze the research questions. The authors present a very detailed description of all variables, materials, procedures, and of the planned analyses. The clarity and degree of methodological detail seem absolutely sufficient to replicate the proposed experimental procedures and analyses.

I have only a two questions that came up when reading the manuscript:

1. In the post-test interview, the scale to assess the actor’s feelings is not symmetrical. There are two options to answer on the “good” side of the scale (“good”, “medium good”) and only one option on the bad feelings side of the scale (“not good”). Was this a deliberate decision? If yes, I would be interested in the reasons for it. I would be concerned that such an asymmetrical scale could lead to distorted answers. Why not use a symmetrical scale with four options (bad, rather bad, rather good, good)? There is a lot of research from the field of moral development that uses such a scale with children in kindergarten and school age.

2. When reading the description of the power analyses, I was surprised about a power of $1-\beta=1$ for the body posture analyses. I am curious if this is correct? In this respect, it would also be helpful to mention the size of the effects from the pilot data. For the behavioral measures, the authors reported the effect size from the pilot study, but they might have missed it for the body posture analyses.

Conclusion

All in all, this is a sound study with an interesting research paradigm that focusses on a relevant research question in order to better understand children's ontogeny of social emotions in the context of unfairness. All procedures sound feasible and I recommend accepting this study, once the abovementioned issues are clarified.

Literature used in this review

Beißert, H. M., Mulvey, K. L., & Killen, M. (2018). Children's Act Evaluation and Emotion Attribution Reasoning Regarding Different Moral Transgressions. *Merrill-Palmer Quarterly*, *64*(2), 195-219.

Malti, T., & Krettenauer, T. (2013). The relation of moral emotion attributions to prosocial and antisocial behavior: A meta-analysis. *Child Development*, *84*(2), 397-412.

Nunner-Winkler, G. (1998). The development of moral understanding and moral motivation. *International Journal of Educational Research*, *27*(7), 587-603.

Nunner-Winkler, G., & Sodian, B. (1988). Children's understanding of moral emotions. *Child development*, 1323-1338.

Appendix C

Editor:

“Three expert reviewers have now assessed the manuscript. All are generally positive about the proposal but note a range of areas that require clarification and possible revision, including consideration of additional literature underpinning the rationale, clarity of the hypotheses, adequacy of the power analyses and controls, and possible consideration of pilot data. All issues appear to be readily addressable in a thorough revision and response.”

1. In the revised version of the manuscript we now consider additional literature as per the reviewers’ suggestions. The relevant passages are the following:

Relatedly, although both younger and older children are aware of the impermissibility of violations of social norms, younger children, between age 4 to 6, have been found to attribute positive emotions to third parties and themselves in hypothetical scenarios involving transgressions against social norms (e.g., when stealing candy from another child), thus behaving as “happy victimizers” (Keller, Lourenço, Malti, & Saalbach, 2003; Krettenauer, Malti, & Sokol, 2008; Nunner-Winkler & Sodian, 1988; Pons, Harris, & Rosnay, 2004). Only older children, by age 7 to 9, more consistently attribute negative emotions to transgressors and themselves in such scenarios (e.g., Keller et al., 2003, p. 6, l. 25 - p. 7, l. 6).

A perhaps critical methodological difference between the study by LoBue et al. (2011), and prior work on children’s social emotions is that children in this study did not cause the negative outcome (i.e. the unfairness) themselves, and rather merely witnessed an adult causing it. Yet, viewing the self, rather than another, as the source of a negative or positive outcome is a central antecedent of social emotions such as guilt, shame and pride (Lewis, 2007; Stipek, Recchia, McClintic, & Lewis, 1992; Vaish, Carpenter, & Tomasello, 2016). For instance, 3-year-olds are more likely to show guilt-like responses when they cause harm to another compared to when no harm is caused or the harm is caused by a third party (Vaish et al., 2016). Moreover, by age 4 to 6, preschoolers attribute negative, rather than positive, emotions to transgressors who *unintentionally* violate social norms (Nunner-Winkler & Sodian, 1988, p. 7, l. 12-22).

2. Furthermore, we have revised the hypotheses to improve their clarity. The revised passages are as follows:

If only older children (by age 7 to 9) respond with negative social emotions to advantageous inequity, children by this age – but not younger children – should show more negative emotions to (4,1) than to (1,1) distributions in the social context (*H1: Behavior and emotion co-emerge*). This hypothesis is supported by a large body of work showing that children by this age, but not before, reject advantageous resource allocations (e.g., Blake & McAuliffe, 2011; Dunham et al., 2018). Alternatively, if younger and older children show similar social emotions in response to advantageous inequity, then children across the entire age range should respond with more negative emotions to (4,1) than to (1,1) distributions in the social context (*H2: Emotion precedes behavior*, p. 9, l. 8-17).

3. In the revised version of the manuscript, we further elaborate on the power analyses:

All models for the analysis of children's change in body posture were calculated with pilot data (N=21 children provided body posture data). As an index of effect size, we report ΔR_m^2 resulting from the difference in model fit between full models including all fixed and random effects, and null models including only the control variables and random effects without the fixed effects of interest (see *Table 1*). R-squared was calculated with the in R-package *MuMin* (Multi-model inference, Barton, 2019), and using the method described by Nakagawa & Schielzeth (2013). These model comparisons yielded an average effect size of $\Delta R_m^2 = .065$ for the inclusion of the fixed effects of interest in Models 1a and 1b and an average effect size $\Delta R_m^2 = .015$ for the inclusion of the fixed effects of interest Model 2a and 2b. Power analyses based on model comparisons were conducted with the R-package *simr* (Version 1.0.3, Green & MacLeod, 2016), and indicated an average power of $1 - \beta = 1$, 95% CI [.9963 1] to detect effects of this magnitude with the specified sample size (p. 24, l. 8-19).

4. We have elected not to collect pilot data from additional participants, because we concluded that all points raised by reviewers could be addressed by elaborating on procedural decisions that are based on already completed piloting.

For instance, reviewers suggested piloting a procedure in which data is collected from both recipients and actors. We have included a clarification as to why we chose to focus only on one child's body posture:

We piloted a version in which data was collected from actors and recipients, however, having both children take turns walking towards the *Kinect* resulted in an increased study duration, which lead to children becoming increasingly inattentive (p. 12, l. 19-22).

In addition, reviewers suggested piloting different resource allocations. We included the following passage in our revised manuscript to clarify that the chosen resource distribution was the result of piloting the procedure with children in the planned age range:

The final choice of these resource distributions was the result of piloting the procedure with children across a wide age range (p. 13, l. 10-11).

In the following section, we respond to further suggestions by reviewers.

Reviewer 1:

1. *“The assumptions are embedded in a sound theoretical and empirical framework. The authors start the introduction by bringing up prior findings and theories on the development of social emotions and inequity aversion. The strong human concern for fairness and inequity aversion are presented as the base of this research. Relevant findings related to these two aspects in adults and children are presented. However, there is no reference to the happy victimizer paradigm (e.g., Nunner-Winkler & Sodian, 1988; Nunner-Winkler, 1998; Malti & Krettenauer, 2013) which would enhance the literature review. Research from this field also demonstrated that although children know and value specific norms (such as fairness) from early age on, they however do feel good when they experience advantageous unfairness themselves (e.g., Beißert, Mulvey & Killen, 2018) and they also expect others to feel good as well when receiving an unfair reward (e.g., Nunner- Winkler, 1998). That is, children do strive for fairness and equity, however at least young children typically do feel good in the case of advantageous fairness. It might be helpful, to include this field of research in the theoretical considerations and relate it to the current research.*

Also, this phenomenon might be relevant for the current hypotheses. Given the happy victimizer phenomenon, it could be expected that children in the age of the planned sample might still be happy victimizers to some extent (that is, they could show positive emotions in the case of advantageous inequity). Research showed that until age 6-9 (or even older), there is still a considerable amount of children who are happy victimizers. In my opinion, the authors' hypotheses are still valid, though. But it would enhance the rationale to include this aspect in the theoretical considerations."

We have included a consideration of additional literature on the "happy victimizer" phenomenon:

Relatedly, although both younger and older children are aware of the impermissibility of violations of social norms, younger children, between age 4 to 6, have been found to attribute positive emotions to third parties and themselves in hypothetical scenarios involving transgressions against social norms (e.g., when stealing candy from another child), thus behaving as "happy victimizers" (Keller, Lourenço, Malti, & Saalbach, 2003; Krettenauer, Malti, & Sokol, 2008; Nunner-Winkler & Sodian, 1988; Pons, Harris, & Rosnay, 2004). Only older children, by age 7 to 9, more consistently attribute negative emotions to transgressors and themselves in such scenarios (e.g., Keller et al., 2003, p. 6, l. 25 - p. 7, l. 6).

2. "In the post-test interview, the scale to assess the actor's feelings is not symmetrical. There are two options to answer on the "good" side of the scale ("good", "medium good") and only one option on the bad feelings side of the scale ("not good"). Was this a deliberate decision? If yes, I would be interested in the reasons for it. I would be concerned that such an asymmetrical scale could lead to distorted answers. Why not use a symmetrical scale with four options (bad, rather bad, rather good, good)? There is a lot of research from the field of moral development that uses such a scale with children in kindergarten and school age."

We thank the reviewer for this suggestion. We agree that a 4-level scale is more suitable to measure children's emotions in the post-test interview and have adapted the procedure accordingly:

In addition, a post-test interview will be conducted. Another unequal (4,1) and equal (1,1) distribution will be created (order identical to the block order during the test phase) and the

actor will be asked for each one “How did you feel when the distribution was like this?” with the options “good”, “rather good”, “rather bad” and “bad” (see Harris, Donnelly, Guz, & Pitt-Watson, 1986, p. 19, l. 25 - p. 20, l. 3).

3. “When reading the description of the power analyses, I was surprised about a power of $1-\beta=1$ for the body posture analyses. I am curious if this is correct? In this respect, it would also be helpful to mention the size of the effects from the pilot data. For the behavioral measures, the authors reported the effect size from the pilot study, but they might have missed it for the body posture analyses.”

We have now added the effect size for the body posture analyses. The relevant passage is the following (please see also our response to the third editorial comment for the full paragraph):

These model comparisons yielded an average effect size of $\Delta R_m^2=.065$ for the inclusion of the fixed effects of interest in Models 1a and 1b and an average effect size $\Delta R_m^2= .015$ for the inclusion of the fixed effects of interest Model 2a and 2b (p. 24, l. 14-16).

Reviewer 2:

1. “The hypotheses presented are logical and well-motivated. However, I thought that the wording of H1 and H2 could be clearer. H1, is over 4 lines of text which makes it difficult to unpack. I did appreciate the Figures of the hypotheses in the supplementary materials which were very helpful depictions of the various hypotheses.”

We have revised the hypotheses to improve their clarity:

If only older children (by age 7 to 9) respond with negative social emotions to advantageous inequity, children by this age – but not younger children – should show more negative emotions to (4,1) than to (1,1) distributions in the social context (*H1: Behavior and emotion co-emerge*). This hypothesis is supported by a large body of work showing that children by this age, but not before, reject advantageous resource allocations (e.g., Blake & McAuliffe, 2011; Dunham et al., 2018). Alternatively, if younger and older children show similar social emotions in response to advantageous inequity, then children across the entire age range should respond with more negative emotions to (4,1) than to (1,1) distributions in the social context (*H2: Emotion precedes behavior*, p. 9, l. 8-17).

2. “I remain somewhat unclear as to the meaning of “vertically flipping” and how it will be carried out. A photo or schematic of how this will actually be carried out would be helpful. “

We thank the reviewer for pointing out that this aspect of the study procedure was unclear. To clarify the procedure, we have included a schematic depiction of the sequence of events during the test trials:

Figure 2. A schematic depiction of the sequence of events during each test trial (an unequal trial is depicted). Initially, two resource boxes contain the same number of resources in both the top (4) and bottom compartments (1). After the actor chooses a resource box for each child or herself and the container, the actor opens her own resource box and distributes the stickers from the top compartment to her game plate. Simultaneously, E1 flips the recipient or container’s resource box, so that instead of 4 stickers the top compartment of the recipient or container’s resource box now contains 1 sticker. The actor opens the recipient or container’s resource box and distributes the stickers from the top compartment to the recipient or container’s game plate (p. 15, l. 8 - p. 16, l. 4).

3. “The manuscript was also missing details on how the dyads will be determined. Will this be random? I also wondered if the authors could consider attaining teacher ratings of the

children in the class and/or how friendly the children within each dyad are to one another. It is my expectation that friendship status is likely to play a role in driving children's decision making in these sorts of contexts. This might be particularly important when examining differences between the social and non-social conditions."

We have added the following passage to the manuscript to clarify how children will be assigned to dyads:

In the social context, actors will participate with recipients of approximately the same age (within a range of +/- 1.5 years age difference) and the same gender (N=96). Children will be recruited from the same kindergarten or school, and randomly assigned to their respective role within the dyad. Other than gender and age there will be no additional criteria for assigning children to dyads (p. 12, l. 15-19).

Since determining children's friendship status with precision might require additional observation and coding, we consider adding friendship status as an additional independent variable to our analysis plan to be beyond the scope of the current study. Yet, we agree with the reviewer that discussing the potential influence of friendship status on children's emotions in the stage 2 submission of our manuscript will likely be warranted.

4. *"In the Analysis section page 19, the Authors state that "trial" will be included as a random effects and "trial" as a control. Is it possible for trial to be included as both a fixed effect and a control variable? I'm wondering if one of these is meant to be "order" or "block"?"*

Yes, including trial as a fixed and random effect is possible, and parallels previous applications of body posture analyses (Hepach, Vaish, & Tomasello, 2017). We clarify this in the revised manuscript as follows:

All models will include random effects for trial, children's school or kindergarten, participant, as well as for children's distance from the *Kinect* camera. In addition, all models will include children's gender, and trial as control variables (based on Hepach et al., 2017, p. 21, l. 14-16).

We agree with the reviewer that examining a potential effect of block order is a relevant addition to our analysis plan. We have added the following section on a supplementary analysis to address this suggestion:

Supplementary Analysis. In order to address the potential influence of block order on children's change in body posture, we will conduct a supplementary analysis by running all models including only the data children provided on the first block (only equal or only unequal trials). The results of this analysis will be considered exploratory and will not be used to directly address the hypotheses (p. 23, l. 1-5).

5. *"I had just one question, how will the Authors identify trials in which the incorrect distribution was given (i.e., will this be completed post hoc via offline coding?)."*

We have revised the passage as follows to clarify our planned method for excluding observations:

To be included in the sample actors (a) need to complete the study (i.e. complete all body posture trials, and the behavioral measure of advantageous inequity aversion), and (b) have at least two valid test trials per within-subjects condition. Trials will be considered invalid (a) if they feature a wrong distribution (e.g., [1,4], instead of [4,1]), which will be determined offline from the video recordings of the test sessions by a blind coder, or (b) if no body posture data is available, either because of equipment failure or because no upright skeleton could be mapped (see also Hepach et al., 2017). Children who are excluded due to these criteria will be replaced until the sample size of N=192 is reached (p. 12, l. 22 - p. 13, l. 4).

6. *"One potential difference between the two conditions that could impact the results is the difference between the likelihood of children in the social condition receiving facial feedback from their partner vs no possibility of facial feedback in the non-social condition. The authors state that children will be told that it is a "quiet" game to reduce the possibility that any verbal information could interfere with the performance which is a strength of the procedure, however, there is still the possibility of children receiving non-verbal feedback. Is there a way that the authors could reduce the possibility of this interfering with/accounting for the results? Perhaps, completing some post-hoc coding of the sessions might be a better option?"*

I also noted above, that another possible confound source of differences between conditions could be actor child's preference for the other child in their dyad. Is there a way that the authors could control for this affecting their results?"

Our study design, in which children can receive direct facial feedback from a peer recipient, is based on previous work, which has examined inequity aversion in young children (e.g., Blake & McAuliffe, 2011; LoBue et al., 2011). This increases the ecological validity of the paradigm and ensures that children are aware that a peer is affected by their decision in the social context. We agree that a comparison of this study's findings to other work with absent recipients and without the opportunity for direct facial feedback will be called for in the stage 2 submission of our manuscript. We have added the following passage to our manuscript to clarify why we elected to test children in dyads:

In one study, Kogut (2012) found that children by fourth grade, but not younger children, report being less satisfied following self-advantaging compared to equal distributions of resources after sharing with another child. One potential reason that children below grade four might not have reported different emotions to self-advantaging compared equal outcomes in this study is that recipients were absent and anonymous peers. In support of this view, young children's fairness-related behavior increases when recipients are present compared to when they are absent (House, Henrich, Brosnan, & Silk, 2012, p. 6, l. 14 - 20).

7. "The Authors present important terms like "social context" and "social emotions" without providing additional information as to what they mean. For example, on page 4 lines 22-28, the authors raise the findings relating to disadvantageous inequity resulting in anger particularly in "nonsocial contexts". Given the broad readership of this outlet, I think that the introduction could be strengthened by providing a (brief) section in which the distinction between social and non-social contexts is provided, as well as what the authors mean when they are referring to "social emotions"."

We thank the reviewer for raising these points. We have revised the passages introducing the terms "social" and "nonsocial context" as follows:

In a nonsocial context, i.e., when the resource distribution affects only the acting child, we expect that children will express a positive emotion (an elevated posture) after receiving 4 rewards and a negative emotion (a lowered posture) after receiving 1 reward (p. 3, l. 9-12).

In a social context, i.e., when the reward distribution affects the acting child and a peer recipient, receiving 4 rewards while a peer receives 1 reward should result in a negative social emotion similar to shame or guilt (lowered posture), whereas a 1-1 fair split should result in a positive emotion similar to pride (elevated posture, p. 3, l. 13-17).

While disadvantageous resource distributions are rejected more often when a social partner receives more, disadvantageously unequal resource are also rejected in nonsocial contexts, i.e. when no recipient's stake is affected by the resource distribution (McAuliffe, Blake, Kim, Wrangham, & Warneken, 2013, Sanfey, Rilling, Aronson, Nystrom, & Cohen, 2003). This suggests that disadvantageous inequity aversion may be motivated by fairness concerns, as well as a nonsocial frustration over receiving less than one could. By contrast, advantageous inequity aversion requires resolving a conflict between nonsocial concerns (for gaining more than a fair share of a resource for oneself) and fairness norms (equitable distribution of resources), as these two motives are not aligned when receiving more than one deserves (Engelmann & Tomasello, 2019, p. 4, l. 9-19).

In addition, we have revised the passages introducing the term “social emotions” as follows:

Social emotions, such as shame, guilt and pride, are theorized to have the evolved function of enabling humans to resolve such conflicts between self-interest and social norms, to the degree that they are internalized forms of an evaluation of oneself and one's behavior as viewed from the perspective of one's social group (e.g., Tomasello & Vaish, 2013, p. 4, l. 18-22).

Despite the relevance of social emotions, such as shame, guilt and pride, to following social norms and overriding ones' self-interest (Tomasello & Vaish, 2013), little empirical work has focused on the development of children's social emotions in response to (advantageous) (un)fairness (p. 6, l. 11-14).

Reviewer 3:

1. *“p. 4 – The mentioning of shame comes quite abruptly – a further sentence as to how the authors link shame to advantageous inequity or indeed the acknowledgment of fairness would be helpful.”*

We have now specified the relation between social emotions and (un)fairness as follows:

Despite the relevance of social emotions, such as shame, guilt and pride, to following social norms and overriding ones' self-interest (Tomasello & Vaish, 2013), little empirical work has focused on the development of children's social emotions in response to (advantageous) (un)fairness (p. 6, l. 11-14).

2. *“Similarly, I would like the authors to elaborate more on why they think shame is the main negative feeling in reaction to an advantageous resource allocation. What about guilt or embarrassment?”*

We have revised our manuscript to include a consideration of shame, guilt and pride:

Social emotions, such as shame, guilt and pride, are theorized to have the evolved function of enabling humans to resolve such conflicts between self-interest and social norms, to the degree that they are internalized forms of an evaluation of oneself and one's behavior as viewed from the perspective of one's social group (e.g., Tomasello & Vaish, 2013, p. 4, l. 18-22).

Although we have limited the stage 1 manuscript to these three emotions, we acknowledge that a further discussion of the results with respect to other emotions, including embarrassment, in the stage 2 submission of our manuscript may be warranted.

3. *“p.6. – I would like the authors to include more information on alternative emotional expressions that might alter body posture (e.g. disappointment) and how they would distinguish alternative negative (or positive) emotions expressed by a similar body posture.”*

We have included the following clarification in our description of the *Kinect* method to measure children's body posture:

While the *Kinect* camera provides a useful tool to examine changes in children's emotions ranging from more positive to more negative, it is important to note that the body posture outcome measures do not objectively differentiate specific emotions (e.g., disappointment from shame), which can only be accomplished by considering the context in which an emotion was expressed (p. 11, l. 23 – p. 12, l. 3).

In addition, we have revised the introduction to clarify that our aim is not to differentiate between different positive emotions that result in elevated body posture and different negative emotions that result in a lower body posture.

In addition, despite extensive work documenting that adults and children often express social emotions through the body – with an elevated body posture signaling positive social emotions, and a lowered body posture indicating negative social emotions (App, McIntosh, Reed, & Hertenstein, 2011; Keltner & Harker, 1998; Kochanska, Gross, Lin, & Nichols, 2002; Lewis, Alessandri, & Sullivan, 1992; Tracy & Matsumoto, 2008; Wallbott, 1998; see Witkower & Tracy, 2018 for a review) – no previous work has examined whether children express social emotions to (un)fairness through their body posture (p. 7, l. 24 – p. 8, l. 5).

We plan to include a further discussion of the results with respect to whether children showed a shame- or guilt-like negative emotion, and pride-like positive emotion, in the stage 2 submission of our manuscript.

4. *“Generally, I would like to encourage the authors to be very careful when generalising from (observable) behaviour to emotional states. For instance, if a study reports that children reject unequal offers it should not be automatically assumed that this overt behaviour is driven by a (covert) negative emotion, unless additional evidence has been presented (e.g. coding of facial expressions, verbal responses, etc.).”*

We thank reviewer 3 for this suggestion. We have revised our manuscript throughout to ensure that the language used reflects whether behavioral or emotional measures were reported or are assessed. For instance, our hypotheses are now specific to children's social emotions, rather than referring to inequity aversion per se:

If only older children (by age 7 to 9) respond with negative social emotions to advantageous inequity, children by this age – but not younger children – should show more negative emotions to (4,1) than to (1,1) distributions in the social context (*HI: Behavior and emotion co-emerge*, p. 9, l. 8 -11).

Moreover, we have ensured that the description of previous research on children’s inequity aversion specifies whether behavioral measures were collected or responses were determined based on additional coding:

While disadvantageous resource distributions are rejected more often when a social partner receives more, disadvantageously unequal resource are also rejected in nonsocial contexts, i.e. when no recipient’s stake is affected by the resource distribution (McAuliffe, Blake, Kim, Wrangham, & Warneken, 2013, Sanfey, Rilling, Aronson, Nystrom, & Cohen, 2003, p. 4, l. 9-13).

In the – to our knowledge – only study, which examined children’s spontaneous facial expressions of emotion in response to (un)fairness in a peer context, LoBue, Nishida, Chiong, DeLoache, & Haidt (2011) found that 3- to 5-year-old children respond with negative emotions to receiving less of a resource than a peer, yet with neutral to positive emotions to receiving more (p. 6, l. 21-25).

5. “A few ideas:

Why do the authors not measure the body posture of the recipient as well as the actor to collect some data on recipients’ emotional response and to investigate whether there is a link between actors’ and recipients’ emotional reactions? I understand that this would not be possible for the non-social condition, but it would add an interesting layer of additional data and analysis. The method could be adjusted in a way that recipients are “involved” in the final allocation decision as well (e.g. making it a 2-step process, 1. Step: Recipient blindly chooses 2 resources boxes out of many, 2. Step: Actor blindly decides on their allocation).

Similarly, why do the authors not extend the study to disadvantageous inequity aversion, i.e. the actor receiving less (1;4). In their introduction the authors mention the limited number of

studies looking at children’s emotional reactions in unequal allocation situations per se. In including a disadvantageous allocation, the authors would be able to support previous findings that these scenarios elicit disappointment.”

We acknowledge that examining whether disadvantageous inequity elicits a lower posture in the current study would provide corroboration for previous work showing that such contexts elicit disappointment. Yet, the nonsocial control context also provides an outcome-neutral control, with which we expect to corroborate previous work showing that children respond with regret and relief to making less and more advantageous choices in nonsocial contexts.

The reason why we do not plan to collect data from the disadvantaged recipient is that we are concerned that this would make the procedure lengthy – because both children would take turns walking toward the Kinect - and thus tax especially young children’s attention span. Our piloting indicated that children of all ages were engaged in the procedure, as it is currently described, for the entire duration of the test sessions, and that the current length of the study was not an issue.

We have included the following passage in the manuscript to clarify why we do not plan to collect data from the recipient:

We piloted a version in which data was collected from actors and recipients, however, having both children take turns walking towards the *Kinect* resulted in an increased study duration, which lead to children becoming increasingly inattentive (p. 12, l. 19-22).

6. *“It might be thoughtful to pilot different resource allocations before final data collection. An allocation of 5:1 or 6:1 might lead to stronger (and therefore presumably more measurable) emotional expressions, especially for the younger age group.”*

We thank the reviewer for this suggestion. Based on our piloting and previous work (e.g., Blake & McAuliffe, 2011; LoBue et al., 2011), we expect that the chosen (4,1) distribution will be sufficiently unequal to elicit negative emotional responses in the planned sample of children. Moreover, the behavioral measure of advantageous inequity aversion provides an outcome-neutral control for whether the value, ratio, and type of resource are able to elicit a rejection of advantageous inequity in the current sample of children, thus allowing us to rule

out potential alternative explanations, if children do not respond with the predicted emotions in the current study. To clarify why we chose the (4,1) distribution in the current study, we added an additional sentence to the revised manuscript:

The final choice of these resource distributions was the result of piloting the procedure with children across a wide age range (p. 13, l. 10-11).

Appendix D

Oxford, June 6th, 2022

Dame Wendy Hall, Editor-in-Chief, and Chris Chambers, Subject Editor Registered Reports
Royal Society Open Science

Dear Professors:

Please find enclosed our Stage 2 Registered Report entitled, “The ontogeny of children’s social emotions in response to (un)fairness,” which received in-principle-acceptance for publication in *Royal Society Open Science* in December 2019. We are grateful for your understanding and the multiple deadline extensions which allowed us to complete data collection during the COVID-19-pandemic.

In this paper we investigated the development of uniquely human emotions in response to unequal distributions of resources. The study is the first to systematically investigate the origin of negative emotions in response to (advantageous) inequity focusing on changes in children’s body posture (measured via a *Kinect* depth sensor imaging camera). The findings of this study shed new light on the emotional mechanisms underlying the human sense of fairness.

On page 12 of our manuscript, we provide a link to a repository on the Open Science Framework, containing both the study data, analysis scripts, as well as the accepted Stage 1 protocol (<https://osf.io/xem8k/>).

We confirm that no data for the pre-registered study was collected prior to the in-principle acceptance. We confirm that the experiment was executed in the manner originally approved with any unforeseen changes in the approved methods and analyses clearly noted. For instance, as we point out in the *Participants*-section, we adjusted our method of recruiting children due to pandemic-related circumstances during the time of data collection.

Our Stage 2 manuscript contains a substantial number of exploratory analyses. These were either suggested by reviewers in response to our Stage 1 submission or the result of internal discussions within our research team. Following the author guidelines for Registered Reports, we have reported these in separate sections for exploratory analyses and have clearly specified that we do not intend to use the results of these analyses to support our hypotheses.

We confirm that we have not altered the Introduction from the approved Stage 1 protocol. In the *Hypotheses*-section, we used the past tense instead of the present tense.

I will serve as the corresponding author for this submission. My co-authors, Katherine McAuliffe, Peter R. Blake, Daniel B. M. Haun, and Robert Hepach all contributed to this research in a substantial way and approved the final version for submission.

Thank you very much. We look forward to hearing from you.

Sincerely,

Stella Gerdemann

Department of Early Child Development
Faculty of Education, Leipzig University
stella.gerdemann@uni-leipzig.de

Katherine McAuliffe
Boston College

Peter R. Blake
Boston University

Daniel B. M. Haun
Max Planck Institute for Evolutionary Anthropology

Robert Hepach
University of Oxford